Proceedings of the 6th Symposium on Advances in Approximate Bayesian Inference, 2024 1–29

# In-Context In-Context Learning with Transformer Neural Processes

**Matthew Ashman**                                        MCA39@CAM.AC.UK
*University of Cambridge*

**Cristiana Diaconu**                                     CDD43@CAM.AC.UK
*University of Cambridge*

**Adrian Weller**                                         AW665@CAM.AC.UK
*University of Cambridge*
*The Alan Turing Institute*

**Richard E. Turner**                                     RET26@CAM.AC.UK
*University of Cambridge*
*Microsoft Research AI for Science*

## Abstract

Neural processes (NPs) are a powerful family of meta-learning models that seek to approximate the posterior predictive map of the ground-truth stochastic process from which each dataset in a meta-dataset is sampled. There are many cases in which practitioners, besides having access to the dataset of interest, may also have access to other datasets that share similarities with it. In this case, integrating these datasets into the NP can improve predictions. We equip NPs with this functionality and describe this paradigm as *in-context in-context learning*. Standard NP architectures, such as the convolutional conditional NP (ConvCNP) or the family of transformer neural processes (TNPs), are not capable of in-context in-context learning, as they are only able to condition on a single dataset. We address this shortcoming by developing the in-context in-context learning pseudo-token TNP (ICICL-TNP). The ICICL-TNP builds on the family of PT-TNPs, which utilise pseudo-token-based transformer architectures to sidestep the quadratic computational complexity associated with regular transformer architectures. Importantly, the ICICL-TNP is capable of conditioning on both sets of datapoints and sets of datasets, enabling it to perform in-context in-context learning. We demonstrate the importance of in-context in-context learning and the effectiveness of the ICICL-TNP in a number of experiments.

## 1. Introduction

Neural processes (NPs) are a broad family of meta-learning models which learn the mapping from sets of observed datapoints to predictive distributions (Foong et al., 2020). They enjoy a number of attractive properties, most notably their ability to effectively model data from different modalities and drawn from complex stochastic process priors. This flexibility makes NPs a popular choice for a wide variety of problem domains, including spatio-temporal modelling, healthcare, and few-shot learning (Jha et al., 2022). A key consideration of NP architectures is the ability to handle an arbitrary number of observed datapoints in an exchangeable fashion. This is achieved through the use of permutation-invariant set functions, which, in NPs, map from the set of observations to some representation space. A popular choice of architecture for these set functions are transformers (Vaswani et al., 2017;

Lee et al., 2019), giving rise to the family of transformer neural processes (TNPs) (Nguyen and Grover, 2022; Kim et al., 2019; Feng et al., 2022).

In the context of meta-learning, practitioners might have access to related datasets that share similarities with the dataset of interest. If these could be integrated into the NP, its predictions would improve. For example, a meta-dataset of PDE-simulated Navier-Stokes equations could be partitioned into smaller sets of datasets with the same Reynolds number (dimensionless quantity that describes fluid flow). Meta-learning a single NP can only recover the predictive distribution of the marginal—or average—stochastic process used to generate the entire meta-dataset. Instead, if we had access to sufficiently many datasets for each possible Reynolds number, a strictly preferable approach would be to meta-learn multiple NPs—one corresponding to each possible stochastic process (e.g.. Reynolds number). While this would help us recover more accurate predictive distributions, it is often infeasible in practice due to either limited data availability for each possible stochastic process or the existence of a large number of possible stochastic processes.

Rather than learning an individual NP for each stochastic process, we propose to learn a single NP that is able to condition on additional datasets that are known to be drawn from the same stochastic process. Intuitively, this can be thought of as amortising the learning of stochastic-process specific NPs, and we refer to this form of meta-learning as *in-context in-context learning*. Constructing a model that is able to condition on sets of datasets, in addition to sets of observations, requires a function that is able to operate on sets of sets. Recently, there has been a string of research into transformer architectures which perform cross-attention between different data modalities (Jaegle et al., 2021; Borgeaud et al., 2022; Zhang and Yan, 2022; Kim et al., 2021; Shen et al., 2024; Xu et al., 2022, 2023). Our key insight is that we can use the same methods to construct a function that operates on a set of sets, providing a model architecture for in-context in-context learning. In doing so, we develop the in-context in-context learning TNP (ICICL-TNP). We highlight our key contributions as follows:

1. We show that in-context in-context learning can lead to substantially improved predictive performance.

2. We develop a new member of TNPs for in-context in-context learning, the ICICL-TNP. At the core of the ICICL-TNP are pseudo-token based transformer architectures, which enable scaling up to large datasets.

3. We provide an empirical investigation into the ICICL-TNP, demonstrating that it: i) does not reduce the performance of a regular pseudo-token TNP (PT-TNP) model when in-context datasets *are not observed*; and ii) improves predictive performance over a regular PT-TNP model when in-context datasets *are observed*.

## 2. Background

Throughout this section, we use the following notation. Let $\mathcal{X} = \mathbb{R}^{D_x}$ and $\mathcal{Y} = \mathbb{R}^{D_y}$ denote the input and output spaces, and let $(\mathbf{x}, \mathbf{y}) \in \mathcal{X} \times \mathcal{Y}$ denote an input-output pair. Let $\mathcal{S} = \bigcup_{N=0}^{\infty} (\mathcal{X} \times \mathcal{Y})^N$ be a collection of all finite data sets, which includes the empty set $\varnothing$. We denote a context and target set with $\mathcal{D}_c, \mathcal{D}_t \in \mathcal{S}$, where $|\mathcal{D}_c| = N_c$ and $|\mathcal{D}_t| = N_t$.

Let $\mathbf{X}_c \in \mathbb{R}^{N_c \times D_x}$, $\mathbf{Y}_c \in \mathbb{R}^{N_c \times D_y}$ be the inputs and corresponding outputs of $\mathcal{D}_c$, with $\mathbf{X}_t \in \mathbb{R}^{N_t \times D_x}$, $\mathbf{Y}_t \in \mathbb{R}^{N_t \times D_y}$ defined analogously. We denote a single task as $\tau = (\mathcal{D}_c, \mathcal{D}_t) = ((\mathbf{X}_c, \mathbf{Y}_c), (\mathbf{X}_t, \mathbf{Y}_t))$. Let $\mathcal{P}(\mathcal{X})$ denote the collection of $\mathcal{Y}$-valued stochastic processes on $\mathcal{X}$. Let $\Theta$ denote the parameter space of predictive distributions over the outputs. Let $\mathcal{Z}$ denote some latent space.

## 2.1. Neural Processes

NPs (Garnelo et al., 2018a,b) are a type of meta-learning model which seek to learn the *posterior prediction map* $\pi_P : \mathcal{S} \to \mathcal{P}(\mathcal{X})$, which maps from context sets $\mathcal{D}_c$ to the posterior predictive distribution over the target outputs $p(\mathbf{Y}_t | \mathbf{X}_t, \mathcal{D}_c)$ under the ground-truth stochastic process $P$. NP architectures generally consist of an *encoder* $e : \mathcal{S} \times \mathcal{X} \to \mathcal{Z}$, which maps from $\mathcal{D}_c$ and $\mathbf{X}_t$ to some latent representation, and a *decoder* $d : \mathcal{X} \times \mathcal{Z} \to \Theta$, which takes the representation and $\mathbf{X}_t$ as inputs and maps to the parameters of the predictive distribution over the target outputs: $p(\mathbf{Y}_t | \mathbf{X}_t, \mathcal{D}_c) = p(\mathbf{Y}_t | d(\mathbf{X}_t, e(\mathbf{X}_t, \mathcal{D}_c)))$. In this work, we limit our attention to conditional NPs (CNPs) (Garnelo et al., 2018a), which factorise the predictive distribution as $p(\mathbf{Y}_t | \mathbf{X}_t, \mathcal{D}_c) = \prod_{n=1}^{N_t} p(\mathbf{y}_{t,n} | d(\mathbf{x}_{t,n}, e(\mathbf{x}_{t,n}, \mathcal{D}_c)))$. CNPs are trained by maximising the posterior predictive likelihood:

$$\mathcal{L}_{\mathrm{ML}} = \mathbb{E}_{p(\tau)} \left[ \sum_{n=1}^{N_t} \log p(\mathbf{y}_{t,n} | d(\mathbf{x}_{t,n}, e(\mathbf{x}_{t,n}, \mathcal{D}_c))) \right]. \tag{1}$$

Here, the expectation is taken over $p(\tau)$. In practice, we often only have access to a finite number of tasks, in which case we can replace the expectation with a Monte-Carlo estimate.

## 2.2. Transformer Neural Processes

Transformers can be understood as permutation-equivariant set functions (Lee et al., 2019). This makes their use in NPs natural, since we require a set function to construct the mapping from context sets to predictive distributions. For the family of TNPs, this is generally achieved by: 1) obtaining an initial token representation for the context points, $\mathbf{Z}_c^0 \in \mathbb{R}^{N_c \times D_z}$, and target input locations, $\mathbf{Z}_t^0 \in \mathbb{R}^{N_t \times D_z}$, using point-wise embeddings; 2) passing the union of the initial context and target tokens, $\mathbf{Z}^0 = [\mathbf{Z}_c^0, \mathbf{Z}_t^0]$, through a transformer-style architecture; and 3) passing the output tokens (corresponding to the target inputs) of the final layer $L$ of the TNP, $\mathbf{Z}_t^L$, through another MLP to obtain the parameters of the predictive distribution $p(\mathbf{Y}_t | \mathbf{X}_t, \mathcal{D}_c)$. We provide a thorough description of the operations used in transformer-style architectures in Appendix B.

The form of transformer-style architecture varies between members of the family of TNPs—we provide a description of several members of the family in Appendix C. Whilst the regular TNP has many advantages over other NP variants, it is hindered by its large computational complexity of $\mathcal{O}\left(N_c^2 + N_c N_t\right)$ given by the interaction of each token with the entire set of context tokens. This limits its application to relatively small datasets. However, the computational complexity of the standard transformer can be reduced through the introduction of *pseudo-tokens*, leading to pseudo-token based transformers. Let $\mathbf{U} \in \mathbb{R}^{M \times D_z}$ denote an initial set of $M \ll N_c$ tokens we refer to as pseudo-tokens. By only allowing tokens to interact with the set of context tokens $\mathbf{Z}_c$ indirectly through the smaller set of pseudo-tokens, we are able to reduce the computational complexity to $\mathcal{O}\left(M N_c + M N_t + M^2\right)$.

We illustrate the architecture for the perceiver-style approach (Jaegle et al., 2021; Feng et al., 2022) and IST-style approach (Lee et al., 2019) of pseudo-token based transformers in Appendix C. We refer to the family of TNPs that use a pseudo-token based transformer architecture as PT-TNPs.

## 3. In-Context In-Context Learning

In this section, we define the paradigm of in-context in-context learning. Our key result is provided in Theorem 1, which informally states that when provided with multiple datasets drawn from the same stochastic process, we can improve the quality of predictions by taking these datasets into account. We then propose a meta-learning model for achieving this in Section 3.2, the ICICL-TNP.

### 3.1. In-Context In-Context Learning for Mixtures of Stochastic Processes

One of the goals of probabilistic meta-learning can be understood as modelling the posterior prediction map $\pi_P : \mathcal{S} \to \mathcal{P}(\mathcal{X})$, where $P$ denotes the ground truth stochastic process over functions mapping from $\mathcal{X}$ to $\mathcal{Y}$ that our datasets are sampled from. Yet, in many cases $P$ is itself a mixture of stochastic processes, and our datasets can be partitioned into samples from each stochastic process mixture. Concretely, we can model each dataset as

$$\xi_i \sim p(\xi), \quad \mathcal{D}_i \sim P(\xi_i) \tag{2}$$

where $P(\xi_i) \in \mathcal{P}(\mathcal{X})$ is a stochastic process defined by the latent variable $\xi_i \in \Xi$ from which dataset $\mathcal{D}_i$ is sampled. Consider the setting in which we have a set of additional datasets $\{\mathcal{D}_j\}$ also drawn from $P(\xi_i)$. Intuitively, the additional datasets provide information about $P(\xi_i)$, the stochastic process from which $\mathcal{D}_i$ is sampled. Conditioning on these additional datasets would therefore reduce our uncertainty of the ground-truth stochastic process from which $\mathcal{D}_i$ is sampled, and can therefore improve our approximation of the prediction map $\pi_{P(\xi_i)}$. This is formalised in the following theorem:

**Theorem 1 (In-context in-context learning)** *Let $\xi_i \sim p(\xi)$, $\mathcal{D}_i, \{\mathcal{D}_j\} \sim P(\xi_i)$. Let $p(\mathbf{y}|\mathbf{x}, \mathcal{D}_i, \xi_i)$ be the marginal posterior distribution of $P(\xi_i)$ given $\mathcal{D}_i$, $p(\mathbf{y}|\mathbf{x}, \mathcal{D}_i, \{\mathcal{D}_j\})$ be the marginal posterior distribution of the stochastic process $P$ given $\mathcal{D}_i$ and $\{\mathcal{D}_j\}$, and $p(\mathbf{y}|\mathbf{x}, \mathcal{D}_i)$ be the marginal posterior distribution of the stochastic process $P$ given $\mathcal{D}_i$. Then,*

$$\mathbb{E}_{\mathcal{D}_i, \{\mathcal{D}_j\}, \xi_i} \left[ \mathrm{KL} \left[ p(\mathbf{y}|\mathbf{x}, \mathcal{D}_i, \xi_i) || p(\mathbf{y}|\mathbf{x}, \mathcal{D}_i, \{\mathcal{D}_j\}) \right] \right] \leq \mathbb{E}_{\mathcal{D}_i, \xi_i} \left[ \mathrm{KL} \left[ p(\mathbf{y}|\mathbf{x}, \mathcal{D}_i, \xi_i) || p(\mathbf{y}|\mathbf{x}, \mathcal{D}_i) \right] \right]. \tag{3}$$

**Sketch of proof** Observe that $\mathbb{E}_{\mathcal{D}_i, \{\mathcal{D}_j\}, \xi_i} \left[ \mathrm{KL} \left[ p(\mathbf{y}|\mathbf{x}, \mathcal{D}_i, \xi_i) || p(\mathbf{y}|\mathbf{x}, \mathcal{D}_i, \{\mathcal{D}_j\}) \right] \right]$ can be expressed as

$$\begin{aligned}
\mathbb{E}_{\mathcal{D}_i, \{\mathcal{D}_j\}, \xi_i} & \left[ \mathrm{KL} \left[ p(\mathbf{y}|\mathbf{x}, \mathcal{D}_i, \xi_i) || p(\mathbf{y}|\mathbf{x}, \mathcal{D}_i, \{\mathcal{D}_j\}) \right] \right] \\
&= -\mathbb{E}_{\mathcal{D}_i, \xi_i} \left[ \mathcal{H} \left( p(\mathbf{y}|\mathbf{x}, \mathcal{D}_i, \xi_i) \right) \right] - \mathbb{E}_{\mathcal{D}_i, \{\mathcal{D}_j\}, \mathbf{y}} \left[ \log p(\mathbf{y}|\mathbf{x}, \mathcal{D}_i, \{\mathcal{D}_j\} \right] \\
&= -\mathbb{E}_{\mathcal{D}_i, \xi_i} \left[ \mathcal{H} \left( p(\mathbf{y}|\mathbf{x}, \mathcal{D}_i, \xi_i) \right) \right] + \mathbb{E}_{\mathcal{D}_i} \left[ \mathbb{E}_{\{\mathcal{D}_j\}|\mathcal{D}_i} \left[ \mathcal{H} \left( p(\mathbf{y}|\mathbf{x}, \mathcal{D}_i, \{\mathcal{D}_j\}) \right) \right] \right] \\
&\leq -\mathbb{E}_{\mathcal{D}_i, \xi_i} \left[ \mathcal{H} \left( p(\mathbf{y}|\mathbf{x}, \mathcal{D}_i, \xi_i) \right) \right] + \mathbb{E}_{\mathcal{D}_i} \left[ \mathcal{H} \left( p(\mathbf{y}|\mathbf{x}, \mathcal{D}_i) \right) \right] \\
&= \mathbb{E}_{\mathcal{D}_i, \xi_i} \left[ \mathrm{KL} \left[ p(\mathbf{y}|\mathbf{x}, \mathcal{D}_i, \xi_i) || p(\mathbf{y}|\mathbf{x}, \mathcal{D}_i) \right] \right]
\end{aligned} \tag{4}$$

The inequality holds as $\mathcal{H}\left(p(\mathbf{y}|\mathbf{x}, \mathcal{D}_i, \{\mathcal{D}_j\})\right) \leq \mathcal{H}\left(p(\mathbf{y}|\mathbf{x}, \mathcal{D}_i)\right) \; \forall \{\mathcal{D}_j\}$, where $\mathcal{H}\left(\cdot\right)$ denotes the entropy. See Appendix A for a detailed proof. ∎

Note that $p(\mathbf{y}_t|\mathbf{x}_t, \mathcal{D}_i, \{\mathcal{D}_j\}, \xi_i) = p(\mathbf{y}_t|\mathbf{x}_t, \mathcal{D}_i, \xi_i)$, as $\mathcal{D}_i$ and $\{\mathcal{D}_j\}$ are conditionally independent given $\xi_i$. Since we do not observe $\xi_i$, Theorem 1 tells us that we should target the predictive distribution $p(\mathbf{y}|\mathbf{x}, \mathcal{D}_i, \{\mathcal{D}_j\})$, instead of $p(\mathbf{y}|\mathbf{x}, \mathcal{D}_i)$. We refer to this form of learning as *in-context in-context learning* (ICICL), and it requires models that are able to condition on a set of datasets, in addition to individual datasets. We refer to the additional set of datasets as the *in-context datasets*, and shall denote this set as $\{\mathcal{D}_{ic,j}\}_{j=1}^{N_{ic}}$, where $N_{ic}$ is the number of in-context datasets, with $|\mathcal{D}_{ic,j}| = N_{ic,j}$.

### 3.2. In-Context In-Context Learning with Transformer Neural Processes

Whilst current TNP architectures are capable of conditioning on individual datasets, they are not capable of conditioning on an arbitrary number of additional in-context datasets in order to approximate the predictive distribution $p(\mathbf{y}|\mathbf{x}, \mathcal{D}_i, \{\mathcal{D}_{ic,j}\}_{j=1}^{N_{ic}})$. Fortunately, the flexibility afforded by MHCA layers makes this extension natural. As the total number of datapoints within each dataset is potentially large, we shall make use of the pseudo-token based architectures discussed in Section 2.2. There exists a number of possible choices of architecture, but unless stated otherwise, we shall use the architecture illustrated in Figure 1, with a description of other choices of architecture provided in Appendix E. We refer to this family of PT-TNPs as ICICL-TNPs.

At a high level, ICICL-TNPs perform the following set of operations: 1) construct initial token representations of each datapoint within each dataset using point-wise encodings; 2) construct a pseudo-token representation for each of the in-context datasets, $\mathbf{U}_{ic,j}$, and context dataset, $\mathbf{U}$; 3) perform cross-attention between the in-context pseudo-tokens and context pseudo-tokens. We note that this architecture forms a valid set function (Zaheer et al., 2017; Wagstaff et al., 2022) on the context datapoints, a valid set function on the set of in-context datasets, and a valid set function on the datapoints within each in-context dataset (see Appendix D for proof). The ICICL-TNP has an asymptotic computational complexity $\mathcal{O}\left(MN_c + MN_t + \sum_{j=1}^{N_{ic}} \left(M_{ic}N_{ic,j} + MM_{ic}\right)\right)$, where $N_{ic}$ denotes the number of in-context datasets, $N_{ic,j}$ denotes the number of datapoints in the $j$-th in-context dataset, and $M_{ic}$ denotes the number of pseudo-tokens used for each in-context dataset. Importantly, this scales linearly in $N_c$, $N_t$, $N_{ic}$ and $N_{ic,j}$, enabling it to scale to both large datasets and many in-context datasets.

The ICICL-TNP is trained in an analogous fashion to CNPs, whereby the model parameters are optimised by maximising the predictive likelihood:

$$\mathcal{L}_{\mathrm{ML}} = \mathbb{E}_{p(\tau)}\left[\sum_{n=1}^{N_t} \log p(\mathbf{y}_{t,n}|d(\mathbf{x}_{t,n}, e(\mathbf{x}_{t,n}, \mathcal{D}_c, \{\mathcal{D}_{ic}\}))\right]. \tag{5}$$

Here, tasks $\tau = (\mathcal{D}_c, \{\mathcal{D}_{ic}\}, \mathcal{D}_t)$ contain additional in-context datasets, which are fed into the encoder.

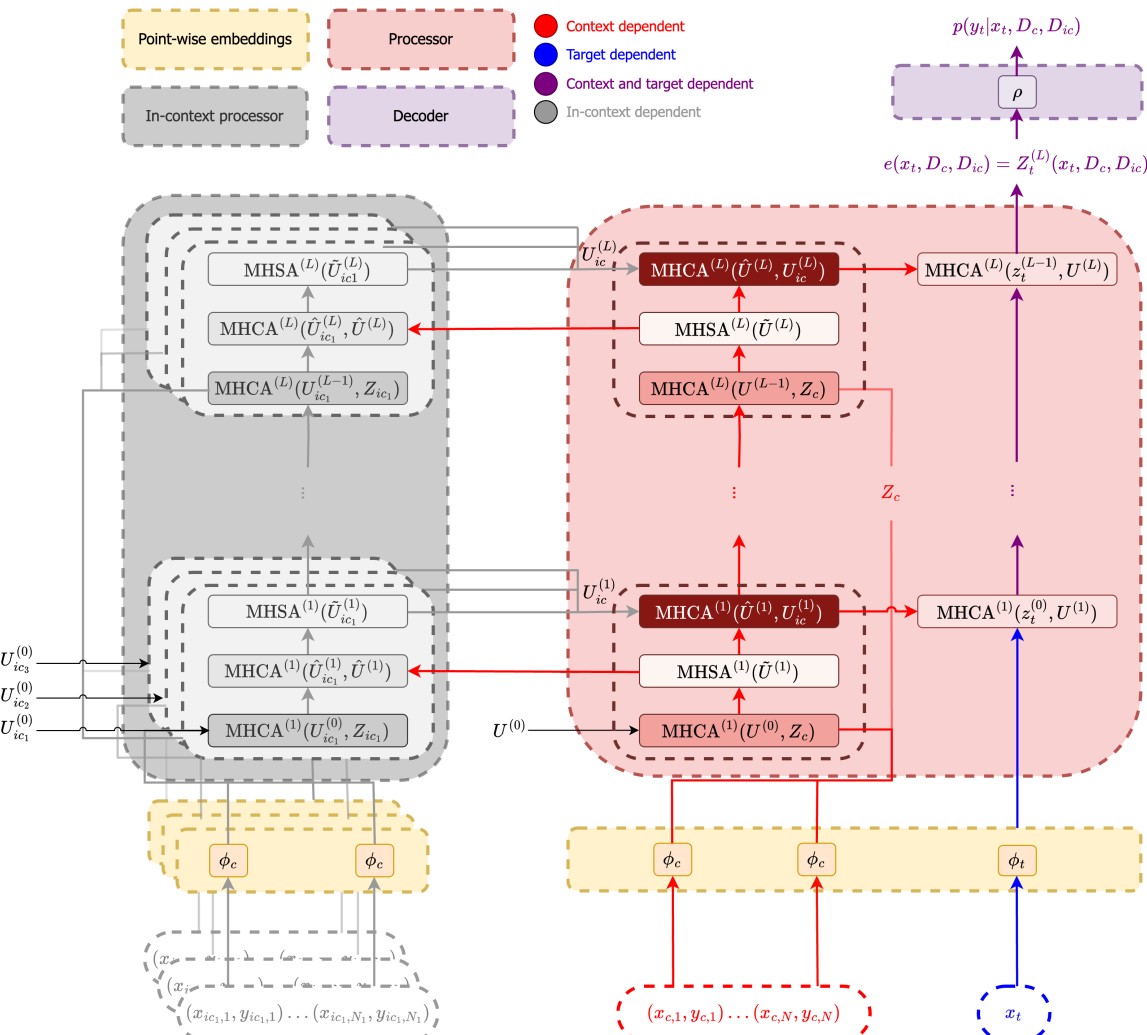

**Figure 1:** A diagram illustrating the ICICL-TNP architecture with three in-context datasets. The point-wise embedding layer is used to get an initial token representation of all datapoints, including the target input location $\mathbf{x}_t$. Then for each layer of the processor, pseudo-token representations for each of the in-context datasets, $\mathbf{U}_{ic}$, and the context dataset, $\mathbf{U}$, are updated through multi-head cross attention (MHCA) operations. The in-context pseudo-tokens $\mathbf{U}_{ic}$ are then modulated by the context pseudo-tokens $\mathbf{U}$, followed by multi-head self attention (MHSA) operations on each set of pseudo-tokens. The in-context pseudo-tokens then modulate the context pseudo-tokens, and finally the context pseudo-tokens modulate the token representation of the target input, $\mathbf{z}_t$. After $L$ layers, the processor outputs the encoder representation $e(\mathbf{x}_t, \mathcal{D}_c, \{\mathcal{D}_{ic,j}\}_{j=1}^{j=N_{ic}})$. Pseudo-code for a forward pass through the ICICL-TNP is provided in Appendix F.

## 4. Related Work

**Cross-Attention Based Architectures in NLP**  Although not described as in-context in-context learning, similar architectures have been proposed for conditioning on additional datasets in NLP. Notably, the retrieval-enhanced transformer (RETRO) (Borgeaud et al., 2022) performs cross-attention between the token representations of additional 'retrieved' texts, demonstrating an improvement in performance over the original transformer architecture (Vaswani et al., 2017). However, the architecture used in RETRO differs substantially from ours. Rather than conditioning on entire datasets, RETRO performs chunking of text to perform next chunk prediction, in which other chunks similar to the previous chunks are used to modulate the predictions. Further, they do not make use of pseudo-token-based transformer architectures to enable scaling to chunks containing more than 64 tokens.

**Cross-Attention Based Architectures for Multi-Task Learning**  A closely related family of transformer architectures are those used for multi-task learning (Zhang and Yan, 2022; Kim et al., 2021; Shen et al., 2024; Xu et al., 2022) and multi-modal learning (Jaegle et al., 2021; Xu et al., 2023). An important difference between this style of architecture and that used for in-context in-context learning is that exchangeability of the in-context datasets is not required, nor desirable. This owes to the fact that multi-output and multi-modal datasets are not independent samples from the same underlying stochastic process. Rather, they can be understood as multiple outputs of the same sample from some stochastic process. The difference is subtle, and in practice is achieved through positional encodings for each output of multi-output data, and different tokenisations of each mode in multi-modal data.

**Conditioning on Exchangeable Datasets in Causal ML**  Finally, the causal structure induction with a supervised approach (CSIvA) model from Ke et al. (2022) also shares similarities with our approach. Similar to the ICICL-TNP, the CSIvA conditions on multiple datasets in an exchangeable manner to make inference. The two models differ significantly elsewhere, however. Whereas we are interested in predictive distributions over outputs, Ke et al. (2022) are interested in inferring causal structure from observational and interventional data. These differences materialise in their use of positional encoding to indicate the identity of the causal node of observations. In a sense, this model bears more resemblance to the multi-task learning architectures, with the additional ability to condition on many instances of samples from the stochastic process in a single forward pass.

## 5. Experiments

In this section, we investigate the performance of the ICICL-TNP. We seek to answer two questions: 1) can the ICICL-TNP recover the performance of a regular PT-TNP when no in-context datasets are provided; 2) how does the performance of the ICICL-TNP vary with the number of in-context datasets provided. In each experiment, we compare the performance of the ICICL-TNP with a regular PT-TNP equivalent. We also provide results for a version of the CNP that supports ICICL (see Appendix G for details of the architecture) alongside the regular CNP. We provide more thorough experimental details in Appendix H, including the choice of model architecture and model training.

## 5.1. Synthetic Regression

We consider a synthetic 1-D regression task using samples drawn from Gaussian processes (GPs) with different kernel types and different kernel hyperparameters. First, we sample either a radial basis function (RBF) or periodic kernel. The kernel hyperparameter $\ell$—corresponding to the lengthscale and period for the RBF and periodic kernel, respectively—is sampled as $\log \ell \sim \mathcal{U}_{[\log 0.25, \log 4]}$. This is shared between the context and in-context datasets. For each context dataset, the number of context points is drawn according to $N_c \sim \mathcal{U}\{1, 64\}$, while the number of target points is set to $N_t = 128$. The context and target inputs are sampled from $x_c \sim \mathcal{U}_{[-2,2]}$ and $x_t \sim \mathcal{U}_{[-4,4]}$. For each such context dataset, we sample $N_{ic}$ in-context datasets with $N_{ic} \sim \mathcal{U}\{0, 5\}$, where the number of datapoints for each in-context dataset is sampled as $N_{ic,j} \sim \mathcal{U}\{64, 128\}$. The in-context inputs are sampled according to $x_{ic} \sim \mathcal{U}_{[-4,4]}$. The observation noise is set to 0.2, and the test set consist of 80,000 datasets.

Table 1 shows that when no in-context information is available, the ICICL-TNP achieves similar performance to that of the PT-TNP (within one standard deviation), and significantly outperforms the CNP. Conditioning on a single in-context dataset significantly improves the predictive performance of the ICICL-TNP, with the improvements in performance plateauing as the number of in-context datasets is increased further. Importantly, these results demonstrate that 1) the ICICL-TNP is able to recover the performance of the regular PT-TNP when no in-context datasets are provided; and 2) the ICICL-TNP is able to perform in-context in-context learning effectively. We compare the predictive distributions of

| Model | Log lik. ($\uparrow$) |
|---|---|
| CNP | $-0.812 \pm 0.005$ |
| PT-TNP | $-0.598 \pm 0.005$ |
| ICICL-TNP (0) | $-0.607 \pm 0.005$ |
| ICICL-TNP (1) | $-0.499 \pm 0.005$ |
| ICICL-TNP (2) | $-0.474 \pm 0.005$ |
| ICICL-TNP (3) | $-0.469 \pm 0.005$ |
| ICICL-TNP (4) | $-0.467 \pm 0.005$ |
| ICICL-TNP (5) | $\mathbf{-0.466 \pm 0.005}$ |

**Table 1:** Comparison of the predictive performance (in terms of test log likelihood) between the CNP, PT-TNP, and the ICICL-TNP with varying number of in-context datasets (indicated within brackets).

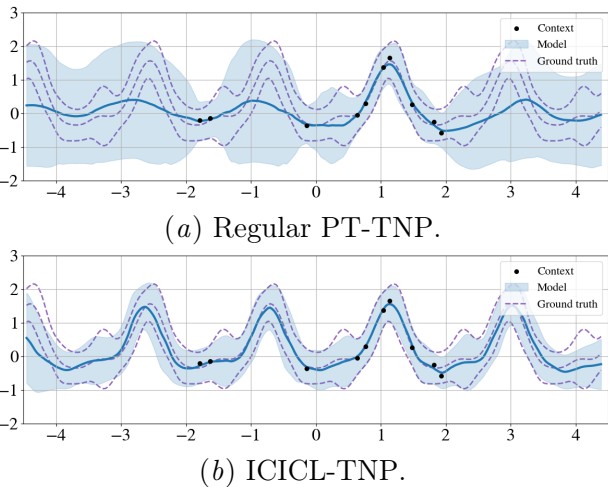

(*a*) Regular PT-TNP.

(*b*) ICICL-TNP.

**Figure 2:** The difference between the predictive distributions of the regular PT-TNP and the ICICL-TNP when conditioning on three in-context datasets with 128 datapoints (not shown here).

the ICICL-TNP and PT-TNP in Figure 2. With only 10 context points, the PT-TNP has insufficient information to infer the periodicity of the underlying stochastic process, so it fails to both extrapolate beyond or interpolate between the data effectively. In contrast, when an in-context dataset drawn from the same periodic kernel is provided, the ICICL-

TNP is able to accurately model the posterior predictive map of the ground-truth periodic stochastic process. Not only does this demonstrate the importance of in-context in-context learning, it also demonstrates the effectiveness of the ICICL-TNP in realising it. Additional experiments are provided in Appendix H.1, illustrating the effect on the predictions of the ICICL-TNP of conditioning on in-context datasets drawn from a different stochastic process.

We also trained an ICICL-CNP model on this task, but we did not observe any benefits from in-context in-context learning. This is in contrast to what we observe using ICICL-TNP on this task, as well as what we observe in Section 5.2. Thus, we hypothesise that the failure of the ICICL-CNP model to perform in-context in-context learning on GP synthetic regression arises from its limited capacity relative to ICICL-TNP and from the difficulty of the task.

**Out-of-distribution (OOD) testing** We also investigated the behaviour of the ICICL-TNP, as compared to PT-TNP, when tested OOD–during testing we sampled the kernel hyperparameter according to $\log \ell \sim \mathcal{U}_{[\log 0.1, \log 0.25] \cup [\log 4, \log 10]}$ for both the context and in-context datasets. Table 2 shows that in-context learning improves predictive performance even if at test time the samples come from stochastic processes the model has not been trained on. This is reflected in Figure 3, where the context datapoints come from a GP with a periodic kernel and $\ell = 6.08$. The ICICL-TNP is better than the PT-TNP at capturing in its uncertainty the slowly-varying variations characteristic to this kernel. We show more examples in Appendix H.1.

| Model | Log lik. ($\uparrow$) |
|---|---|
| CNP | $-0.880 \pm 0.006$ |
| PT-TNP | $-0.798 \pm 0.007$ |
| ICICL-TNP (0) | $-0.783 \pm 0.007$ |
| ICICL-TNP (1) | $-0.721 \pm 0.006$ |
| ICICL-TNP (2) | $-0.702 \pm 0.006$ |
| ICICL-TNP (3) | $\mathbf{-0.700 \pm 0.006}$ |

**Table 2:** Comparison of the predictive performance (in terms of test log likelihood) when tested OOD between the CNP, PT-TNP, and the ICICL-TNP with varying number of in-context datasets (indicated within brackets).

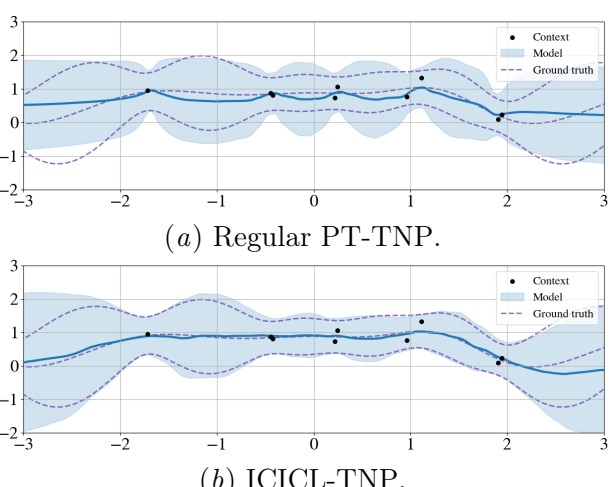

(*a*) Regular PT-TNP.

(*b*) ICICL-TNP.

**Figure 3:** The difference between the predictive distributions when tested OOD of the regular PT-TNP and the ICICL-TNP when conditioning on three in-context datasets. The context datapoints come from a GP with a periodic kernel with $\ell = 6.08$.

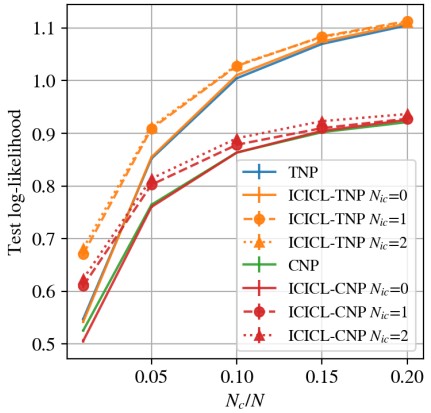

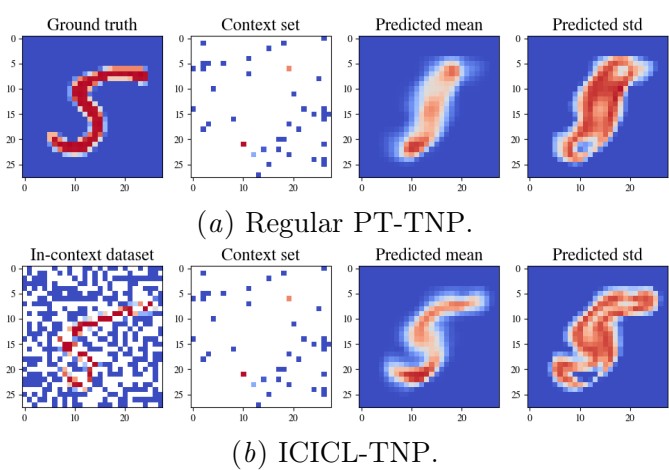

(a) Regular PT-TNP.

(b) ICICL-TNP.

**Figure 4:** A comparison between the predictive performance of the ICICL-TNP, regular PT-TNP, ICICL-CNP and CNP as the proportion of context datapoints $N_c/N$ varies in the MNIST in-painting experiment.

**Figure 5:** A comparison between the predictive distribution of the ICICL-TNP and the regular PT-TNP when conditioning on the context and in-context dataset shown.

## 5.2. Image Completion

We consider an image in-painting experiment using the MNIST dataset (LeCun et al., 1998). Each MNIST image can be interpreted as spatial regression of pixel values $\mathbf{y}_n \in \mathbb{R}$ given a 2-D pixel location $\mathbf{x}_n \in \mathbb{R}^2$. We construct context datasets by sampling the number of pixels according to $N_c \sim \mathcal{U}\{N/100, N/5\}$, where $N = 784$ denotes the total number of pixels in each image, and set $N_t = N - N_c$. For each context dataset, we sample in-context datasets from other images of the same label with the number of in-context datasets sampled as $N_{ic} \sim \mathcal{U}\{0, 3\}$, and sample the number of datapoints for each in-context dataset as $N_{ic,c} \sim \mathcal{U}\{N/100, N/2\}$. Figure 4 compares the predictive performance of the regular PT-TNP with the ICICL-TNP as $N_{ic}$ and $N_c/N$ vary. We also provide results for the ICICL-CNP and CNP.

We observe that the ICICL-TNP recovers the performance of the PT-TNP when conditioning on no in-context datasets, and that, as expected, the gains in performance of the ICICL-TNP plateau with increasing $N_{ic}$. In Figure 5, we compare the predictive distributions of the regular TNP and ICICL-TNP for a single dataset. As very few pixels are observed in the context dataset, the regular TNP is unconfident in its predictions, as the context dataset could correspond to datasets sampled from several MNIST digits. With the addition of an in-context dataset, the ICICL-TNP is able to infer pixel values much closer to the ground truth. We provide additional comparisons in Appendix H.2, showcasing examples where the in-context dataset is drawn from a different stochastic process to the context dataset.

### 5.3. Environmental Data

In this experiment, we model a real-world environmental regression problem derived from ERA5 (Copernicus Climate Change Service, 2020). We consider the surface air temperature across both space and time within central Europe (latitude / longitude range of $[42°, 53°]$ / $[8°, 28°]$). During training, datasets spanning 18 hours (one sample every six hours) and $5°$ (with a resolution of $0.5°$) across each axis are sampled. In-context datasets are obtained from the same region, but non-overlapping regions in time. Once the spatio-temporal location has been sampled, the number of context points is sampled as $N_c \sim \mathcal{U}\{N/100, N/3\}$, where $N = 300$ denotes the maximum number of measurements in a single dataset. The number of in-context datasets is sampled as $N_{ic} \sim \mathcal{U}\{0, 2\}$, with the number of datapoints within each in-context dataset sampled as $N_{ic,j} \sim \mathcal{U}\{N/100, N/5\}$. The number of target points is set to $N_t = N - N_c$. We train on data from the first six months of 2019, and test on the latter six months. Table 3 compares the predictive performance of the ICICL-TNP with the regular PT-TNP. As with the previous two experiments, the ICICL-TNP is able to recover the performance of the PT-TNP with no in-context datasets, and outperforms the PT-TNP when in-context datasets are provided.

Given that in this scenario the distinction between the context and the in-context data is not as clear as in some of the other experiments—a result of the experimental setup—we also include in Appendix H.3 an additional baseline. This consists of a PT-TNP model trained and tested on larger context datasets that result from merging the original context datasets with the in-context datasets (i.e. $\mathcal{D}'_c = \mathcal{D}_c \cup \{\mathcal{D}_{ic,j}\}_{j=1}^{N_{ic}}$). The results are shown in Table 5. We find that merging the in-context datasets with the context dataset does not lead to any performance improvement, likely due to the difficulty in learning complex relationships across long periods of time.

| Model | Test Log-Likelihood |
|---|---|
| PT-TNP | $1.15 \pm 0.01$ |
| ICICL-TNP (0) | $1.15 \pm 0.01$ |
| ICICL-TNP (1) | $1.18 \pm 0.01$ |
| ICICL-TNP (2) | $1.19 \pm 0.01$ |

**Table 3:** Comparison of the test log-likelihood on the environmental data for the PT-TNP and ICICL-TNP with varying number of in-context datasets (indicated within brackets).

### 6. Conclusion

We have introduced the paradigm of in-context in-context learning, in which multiple datasets are used for predictive inference on a dataset drawn from the same stochastic process. We developed a NP for performing in-context in-context learning, the ICICL-TNP, which utilises pseudo-token-based transformer architectures for the computationally efficient handling of both context and in-context datasets with potentially many datapoints in each. Through a set of synthetic and real-world experiments, we demonstrate the importance of in-context in-context learning and the effectiveness of the ICICL-TNP in performing it. Further, we demonstrate that the ICICL-TNP is able to recover the performance of an equivalent

PT-TNP when no in-context datasets are provided. A requirement of in-context in-context learning generally is for practitioners to be able to identify datasets drawn from the same stochastic process, which in turn limits the applicability of the ICICL-TNP. Nonetheless, we believe that by establishing the benefit of in-context in-context learning, new machine learning applications can be developed that utilise it. For example, one can envision developing image generation tools which can be provided with additional images that the user wishes the generated sample to be similar to. We are excited by the abundance of potential directions in which to take this research, and look forward to exploring them in future work.

## Acknowledgments

CD is supported by the Cambridge Trust Scholarship. AW acknowledges support from a Turing AI fellowship under grant EP/V025279/1 and the Leverhulme Trust via CFI. RET is supported by gifts from Google, Amazon, ARM, Improbable and EPSRC grant EP/T005386/1.

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

## Appendix A. Proof of Theorem 1

In this section we give a detailed proof of Theorem 1, which we repeat here for completeness.

**Theorem 1 (In-context in-context learning)** *Let $\xi_i \sim p(\xi)$, $\mathcal{D}_i, \{\mathcal{D}_j\} \sim P(\xi_i)$. Let $p(\mathbf{y}|\mathbf{x}, \mathcal{D}_i, \xi_i)$ be the marginal posterior distribution of $P(\xi_i)$ given $\mathcal{D}_i$, $p(\mathbf{y}|\mathbf{x}, \mathcal{D}_i, \{\mathcal{D}_j\})$ be the marginal posterior distribution of the stochastic process $P$ given $\mathcal{D}_i$ and $\{\mathcal{D}_j\}$, and $p(\mathbf{y}|\mathbf{x}, \mathcal{D}_i)$ be the marginal posterior distribution of the stochastic process $P$ given $\mathcal{D}_i$. Then,*

$$\mathbb{E}_{\mathcal{D}_i, \{\mathcal{D}_j\}, \xi_i} \left[ \text{KL} \left[ p(\mathbf{y}|\mathbf{x}, \mathcal{D}_i, \xi_i) || p(\mathbf{y}|\mathbf{x}, \mathcal{D}_i, \{\mathcal{D}_j\}) \right] \right] \le \mathbb{E}_{\mathcal{D}_i, \xi_i} \left[ \text{KL} \left[ p(\mathbf{y}|\mathbf{x}, \mathcal{D}_i, \xi_i) || p(\mathbf{y}|\mathbf{x}, \mathcal{D}_i) \right] \right].$$

**Proof** First, observe that both terms of the inequality can be split into two parts:

$$\mathbb{E}_{\mathcal{D}_i, \{\mathcal{D}_j\}, \xi_i} \left[ \text{KL} \left[ p(\mathbf{y}|\mathbf{x}, \mathcal{D}_i, \xi_i) || p(\mathbf{y}|\mathbf{x}, \mathcal{D}_i, \{\mathcal{D}_j\}) \right] \right]$$

$$= \mathbb{E}_{\mathcal{D}_i, \{\mathcal{D}_j\}, \xi_i} \left[ \int \underbrace{p(\mathbf{y}|\mathbf{x}, \mathcal{D}_i, \xi_i) \log p(\mathbf{y}|\mathbf{x}, \mathcal{D}_i, \xi_i)}_{\text{term 1}} \, d\mathbf{y} - \underbrace{p(\mathbf{y}|\mathbf{x}, \mathcal{D}_i, \xi_i) \log p(\mathbf{y}|\mathbf{x}, \mathcal{D}_i, \{\mathcal{D}_j\})}_{\text{term 2}} \, d\mathbf{y} \right] \tag{6}$$

and

$$\mathbb{E}_{\mathcal{D}_i, \xi_i} \left[ \text{KL} \left[ p(\mathbf{y}|\mathbf{x}, \mathcal{D}_i, \xi_i) || p(\mathbf{y}|\mathbf{x}, \mathcal{D}_i) \right] \right]$$

$$= \mathbb{E}_{\mathcal{D}_i, \xi_i} \left[ \int \underbrace{p(\mathbf{y}|\mathbf{x}, \mathcal{D}_i, \xi_i) \log p(\mathbf{y}|\mathbf{x}, \mathcal{D}_i, \xi_i)}_{\text{term 1}} \, d\mathbf{y} - \underbrace{p(\mathbf{y}|\mathbf{x}, \mathcal{D}_i, \xi_i) \log p(\mathbf{y}|\mathbf{x}, \mathcal{D}_i)}_{\text{term 2}} \, d\mathbf{y} \right]. \tag{7}$$

We first show that term 1 is the same for both sides of the inequality:

$$\mathbb{E}_{\mathcal{D}_i, \{\mathcal{D}_j\}, \xi_i} \left[ \int p(\mathbf{y}|\mathbf{x}, \mathcal{D}_i, \xi_i) \log p(\mathbf{y}|\mathbf{x}, \mathcal{D}_i, \xi_i) d\mathbf{y} \right]$$

$$= \mathbb{E}_{\xi_i \sim p(\xi), \mathcal{D}_i \sim P(\xi_i), \{\mathcal{D}_j\} \sim P(\xi_i)} \left[ \int p(\mathbf{y}|\mathbf{x}, \mathcal{D}_i, \xi_i) \log p_(\mathbf{y}|\mathbf{x}, \mathcal{D}_i, \xi_i) d\mathbf{y} \right]$$

$$= \mathbb{E}_{\xi_i \sim p(\xi), \mathcal{D}_i \sim P(\xi_i)} \left[ \int p(\mathbf{y}|\mathbf{x}, \mathcal{D}_i, \xi_i) \log p(\mathbf{y}|\mathbf{x}, \mathcal{D}_i, \xi_i) d\mathbf{y} \right]$$

$$= \mathbb{E}_{\mathcal{D}_i, \xi_i} \left[ \int p(\mathbf{y}|\mathbf{x}, \mathcal{D}_i, \xi_i) \log p(\mathbf{y}|\mathbf{x}, \mathcal{D}_i, \xi_i) d\mathbf{y} \right]$$

$$= -\mathbb{E}_{\mathcal{D}_i, \xi_i} \left[ \mathcal{H} \left( p(\mathbf{y}|\mathbf{x}, \mathcal{D}_i, \xi_i) \right) \right] \tag{8}$$

where to go from the second to the third line we leveraged the fact that the term within the expectation does not depend on $\{\mathcal{D}_j\}$, and can, hence, be integrated out. Thus, the elements of the inequality only differ in term 2. Consider term 2 in Equation 6:

$$\mathbb{E}_{\mathcal{D}_i, \{\mathcal{D}_j\}, \xi_i} \left[ \int p(\mathbf{y}|\mathbf{x}, \mathcal{D}_i, \xi_i) \log p(\mathbf{y}|\mathbf{x}, \mathcal{D}_i, \{\mathcal{D}_j\}) d\mathbf{y} \right]$$

$$= \mathbb{E}_{\mathcal{D}_i, \{\mathcal{D}_j\}} \left[ \mathbb{E}_{\xi_i \sim p(\xi|\mathcal{D}_i, \{\mathcal{D}_j\})} \left[ \int p(\mathbf{y}|\mathbf{x}, \mathcal{D}_i, \xi_i) \log p(\mathbf{y}|\mathbf{x}, \mathcal{D}_i, \{\mathcal{D}_j\}) d\mathbf{y} \right] \right]$$

Note that $p(\mathbf{y}|\mathbf{x}, \mathcal{D}_i, \xi_i) = p(\mathbf{y}|\mathbf{x}, \mathcal{D}_i, \{\mathcal{D}_j\}, \xi_i)$—considering that $\mathcal{D}_i$ and $\{\mathcal{D}_j\}$ are conditionally independent given $\xi_i$. Informally, this means that additionally conditioning on the

in-context datasets $\{\mathcal{D}_j\}$ does not give us any extra information, given that we are already considering the marginal posterior of $P(\xi_i)$. Hence,

$$\mathbb{E}_{\mathcal{D}_i,\{\mathcal{D}_j\},\xi_i}\left[\int p(\mathbf{y}|\mathbf{x},\mathcal{D}_i,\xi_i)\log p(\mathbf{y}|\mathbf{x},\mathcal{D}_i,\{\mathcal{D}_j\})d\mathbf{y}\right]$$

$$= \mathbb{E}_{\mathcal{D}_i,\{\mathcal{D}_j\}}\left[\mathbb{E}_{\xi_i\sim p(\xi|\mathcal{D}_i,\{\mathcal{D}_j\})}\left[\int p(\mathbf{y}|\mathbf{x},\mathcal{D}_i,\{\mathcal{D}_j\},\xi_i)\log p(\mathbf{y}|\mathbf{x},\mathcal{D}_i,\{\mathcal{D}_j\})d\mathbf{y}\right]\right]$$

$$= \mathbb{E}_{\mathcal{D}_i,\{\mathcal{D}_j\}}\left[\int p(\mathbf{y}|\mathbf{x},\mathcal{D}_i,\{\mathcal{D}_j\})\log p(\mathbf{y}|\mathbf{x},\mathcal{D}_i,\{\mathcal{D}_j\})d\mathbf{y}\right]$$

$$= -\mathbb{E}_{\mathcal{D}_i,\{\mathcal{D}_j\}}\left[\mathcal{H}\left(p(\mathbf{y}|\mathbf{x},\mathcal{D}_i,\{\mathcal{D}_j\})\right)\right]$$

$$= -\mathbb{E}_{\mathcal{D}_i}\left[\mathbb{E}_{\{\mathcal{D}_j\}|\mathcal{D}_i}\left[\mathcal{H}\left(p(\mathbf{y}|\mathbf{x},\mathcal{D}_i,\{\mathcal{D}_j\})\right)\right]\right] \tag{9}$$

where we used Fubini's theorem to go from the second to the third line. Similarly, for term 2 of Equation 7 we have:

$$\mathbb{E}_{\mathcal{D}_i,\xi_i}\left[\int p(\mathbf{y}|\mathbf{x},\mathcal{D}_i,\xi_i)\log p(\mathbf{y}|\mathbf{x},\mathcal{D}_i)d\mathbf{y}\right]$$

$$= \mathbb{E}_{\mathcal{D}_i}\left[\mathbb{E}_{\xi_i\sim p(\xi|\mathcal{D}_i)}\left[\int p(\mathbf{y}|\mathbf{x},\mathcal{D}_i,\xi_i)\log p(\mathbf{y}|\mathbf{x},\mathcal{D}_i)d\mathbf{y}\right]\right]$$

$$= \mathbb{E}_{\mathcal{D}_i}\left[\int p(\mathbf{y}|\mathbf{x},\mathcal{D}_i)\log p(\mathbf{y}|\mathbf{x},\mathcal{D}_i)d\mathbf{y}\right]$$

$$= -\mathbb{E}_{\mathcal{D}_i}\left[\mathcal{H}\left(p(\mathbf{y}|\mathbf{x},\mathcal{D}_i)\right)\right]. \tag{10}$$

Putting the results from Equation 8 and Equation 9 together we obtain

$$\mathbb{E}_{\mathcal{D}_i,\{\mathcal{D}_j\},\xi_i}\left[\mathrm{KL}\left[p(\mathbf{y}|\mathbf{x},\mathcal{D}_i,\{\mathcal{D}_j\},\xi_i)||p(\mathbf{y}|\mathbf{x},\mathcal{D}_i)\right]\right] \tag{11}$$

$$= -\mathbb{E}_{\mathcal{D}_i,\xi_i}\left[\mathcal{H}\left(p(\mathbf{y}|\mathbf{x},\mathcal{D}_i,\xi_i)\right)\right] + \mathbb{E}_{\mathcal{D}_i}\left[\mathbb{E}_{\{\mathcal{D}_j\}|\mathcal{D}_i}\left[\mathcal{H}\left(p(\mathbf{y}|\mathbf{x},\mathcal{D}_i,\{\mathcal{D}_j\})\right)\right]\right], \tag{12}$$

and putting Equation 8 and Equation 10 together we have

$$\mathbb{E}_{\mathcal{D}_i,\xi_i}\left[\mathrm{KL}\left[p(\mathbf{y}|\mathbf{x},\mathcal{D}_i,\xi_i)||p(\mathbf{y}|\mathbf{x},\mathcal{D}_i)\right]\right] \tag{13}$$

$$= -\mathbb{E}_{\mathcal{D}_i,\xi_i}\left[\mathcal{H}\left(p(\mathbf{y}|\mathbf{x},\mathcal{D}_i,\xi_i)\right)\right] + \mathbb{E}_{\mathcal{D}_i}\left[\mathcal{H}\left(p(\mathbf{y}|\mathbf{x},\mathcal{D}_i)\right)\right]. \tag{14}$$

Since conditioning reduces entropy $\mathcal{H}\left(p(\mathbf{y}|\mathbf{x},\mathcal{D}_i,\{\mathcal{D}_j\})\leq\mathcal{H}\left(p(\mathbf{y}|\mathbf{x},\mathcal{D}_i)\right)\;\forall\{\mathcal{D}_j\}\sim P(\xi_i)$, we have

$$\mathbb{E}_{\mathcal{D}_i}\left[\mathbb{E}_{\{\mathcal{D}_j\}|\mathcal{D}_i}\left[\mathcal{H}\left(p(\mathbf{y}|\mathbf{x},\mathcal{D}_i,\{\mathcal{D}_j\})\right)\right]\right]\leq\mathbb{E}_{\mathcal{D}_i}\left[\mathbb{E}_{\{\mathcal{D}_j\}|\mathcal{D}_i}\left[\mathcal{H}\left(p(\mathbf{y}|\mathbf{x},\mathcal{D}_i)\right)\right]\right]=\mathbb{E}_{\mathcal{D}_i}\left[\mathcal{H}\left(p(\mathbf{y}|\mathbf{x},\mathcal{D}_i)\right)\right],$$

which leads to the result in Theorem 1. ■

## Appendix B. Defining the MHSA and MHCA Operations

Let $\mathbf{Z}^\ell\in\mathbb{R}^{N\times D_z}$ denote the input set to the $\ell$-th MHSA operation. The MHSA operation updates the $n^{\text{th}}$ token $\mathbf{z}_n^\ell$ as

$$\tilde{\mathbf{z}}_n^\ell=\mathrm{cat}\left(\left\{\textstyle\sum_{m=1}^N\alpha_h^\ell(\mathbf{z}_n^\ell,\mathbf{z}_m^\ell)\mathbf{z}_m^{\ell\,T}\mathbf{W}_{V,h}^\ell\right\}_{h=1}^{H^\ell}\right)\mathbf{W}_O^\ell \tag{15}$$

where cat denotes the concatenation operation across the last dimension. Here, $\mathbf{W}_{V,h}^\ell \in \mathbb{R}^{D_z \times D_V}$ and $\mathbf{W}_O^\ell \in \mathbb{R}^{H^\ell D_V \times D_z}$ are the value and projection weight matrices, where $H^\ell$ denotes the number of 'heads' in layer $\ell$. Note that permutation equivariance is achieved through the permutation invariant summation operator. As this is the only mechanism through which the tokens interact with each other, permutation equivariance for the overall model is ensured. The attention mechanism, $\alpha_h^\ell$, is implemented as

$$\alpha_h^\ell(\mathbf{z}_n^\ell, \mathbf{z}_m^\ell) = \frac{e^{\mathbf{z}_n^{\ell\,T}\mathbf{W}_{Q,h}^\ell[\mathbf{W}_{K,h}^\ell]^T \mathbf{z}_m^\ell}}{\sum_{m=1}^N e^{\mathbf{z}_n^{\ell\,T}\mathbf{W}_{Q,h}^\ell[\mathbf{W}_{K,h}^\ell]^T \mathbf{z}_m^\ell}} \tag{16}$$

where $\mathbf{W}_{Q,h}^\ell \in \mathbb{R}^{D_z \times D_{QK}}$ and $\mathbf{W}_{K,h}^\ell \in \mathbb{R}^{D_z \times D_{QK}}$ are the query and key weight matrices. The softmax-normalisation ensures that $\sum_{m=1}^N \alpha_h^\ell(\mathbf{z}_n^\ell, \mathbf{z}_m^\ell) = 1 \; \forall n, h, \ell$.

Often, conditional independencies amongst the set of tokens—in the sense that the set $\{\mathbf{z}_n^\ell\}_{\ell=1}^{\ell=L}$ do not depend on the set $\{\mathbf{z}_m^\ell\}_{\ell=0}^{\ell=L}$ for some $n$, $m \in \{1, \ldots, N\}$—are desirable. This is typically achieved through masking, such that the pre-softmax activations are replaced by $\tilde{\alpha}_h^\ell$, where

$$\tilde{\alpha}_h^\ell(\mathbf{z}_n^\ell, \mathbf{z}_m^\ell) = \begin{cases} -\infty, & m \in A(n). \\ \mathbf{z}_n^{\ell\,T}\mathbf{W}_{Q,h}^\ell\left[\mathbf{W}_{K,h}^\ell\right]^T \mathbf{z}_m^\ell, & \text{otherwise.} \end{cases} \tag{17}$$

Here, $A(n) \subseteq \mathbb{N}_{\leq N}$ indexes the set of tokens we wish to make the update for token $\mathbf{z}_n^\ell$ independent of. If $A(n) = A$ (i.e. the same set of tokens are conditioned on for every $n$) then in practice it is more computationally efficient to use MHCA operations together with MHSA operations than it is to directly compute Equation 15. An MHCA operation uses the subset of tokens $\{\mathbf{z}_m^\ell | m \in A\}$ to update the complementary set of tokens $\{\mathbf{z}_n^\ell | n \in A^c\}$ in a computationally efficient manner:

$$\tilde{\mathbf{z}}_n^\ell = \text{cat}\left(\left\{\sum_{m \in A} \alpha_h^\ell(\mathbf{z}_n^\ell, \mathbf{z}_m^\ell)\mathbf{z}_m^{\ell\,T}\mathbf{W}_{V,h}^\ell\right\}_{h=1}^{H^\ell}\right)\mathbf{W}_O^\ell \qquad \forall n \in A^c. \tag{18}$$

For $N$ tokens that solely depend on a subset of $N_1$ tokens, the computational complexity is reduced from $\mathcal{O}\left(N^2\right)$ using masked-MHSA operations to $\mathcal{O}\left(NN_1\right)$ using MHCA operations.

## Appendix C. Some TNP Architectures

There exist a number of architectures used in different members of the TNP. We provide diagramatic illustrations of the following: the attentive NP (ANP) of Kim et al. (2019) in Figure 6; the TNP of Nguyen and Grover (2022) in Figure 7; the latent-bottlenecked ANP (LBANP) of Feng et al. (2022) in Figure 8; and the induced set transformer (IST)-style (Lee et al., 2019) TNP in Figure 9.

## Appendix D. Set Functions and Set of Sets Functions

In this section we utilise the Deepset result of Zaheer et al. (2017), which states that any permutation invariant function on the set $\{z_n\}_{n=1}^N$ can be expressed as

$$f(z_1, z_2, \ldots, z_N) = \rho\left(\sum_{n=1}^N \phi(z_n)\right) \tag{19}$$

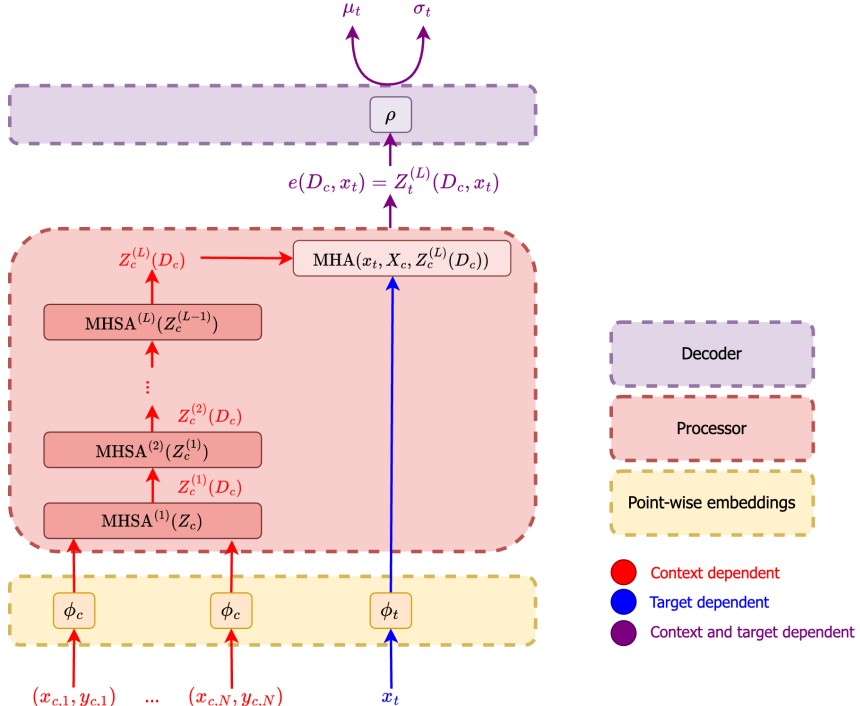

**Figure 6:** A diagram illustrating the architecture of the ANP (Kim et al., 2019).

for continuous functions $\rho$ and $\phi$. As shown in (Lee et al., 2019), the Deepset formulation is a special case of pseudo-token based transformers operating on sets. It is therefore sufficient to consider only this Deepset formulation.

Let $N_{ic}$ be the number of in-context datasets, each with $N_1, N_2, \ldots, N_{N_{ic}}$ datapoints. Each in-context dataset is embedded into a latent representation through a valid set function of the form

$$f(z_{n,1}, z_{n,2}, \ldots, z_{n,N_n}) = \rho_n\left(\sum_{j=1}^{N_n} \phi_n(z_{n,j})\right)$$

for all $n \in \{1, 2, \ldots, N_{ic}\}$. When combining the representation of all of the in-context datasets, we also obtain a valid set function

$$f\left(\{z_{1,1}, z_{1,2}, \ldots, z_{1,N_1}\}, \ldots, \{z_{N_{ic},1}, z_{N_{ic},2}, \ldots, z_{N_{ic},N_{N_{ic}}}\}\right) =$$
$$f\left(\rho_1\left(\sum_{j=1}^{N_1} \phi_1(z_{1,j})\right), \ldots, \rho_{N_{ic}}\left(\sum_{j=1}^{N_{N_{ic}}} \phi_{N_{ic}}(z_{N_{ic},j})\right)\right) =$$
$$\rho\left(\sum_{n=1}^{N_{ic}} \phi\left(\rho_n\left(\sum_{j=1}^{N_n} \phi_n(z_{n,j})\right)\right)\right).$$

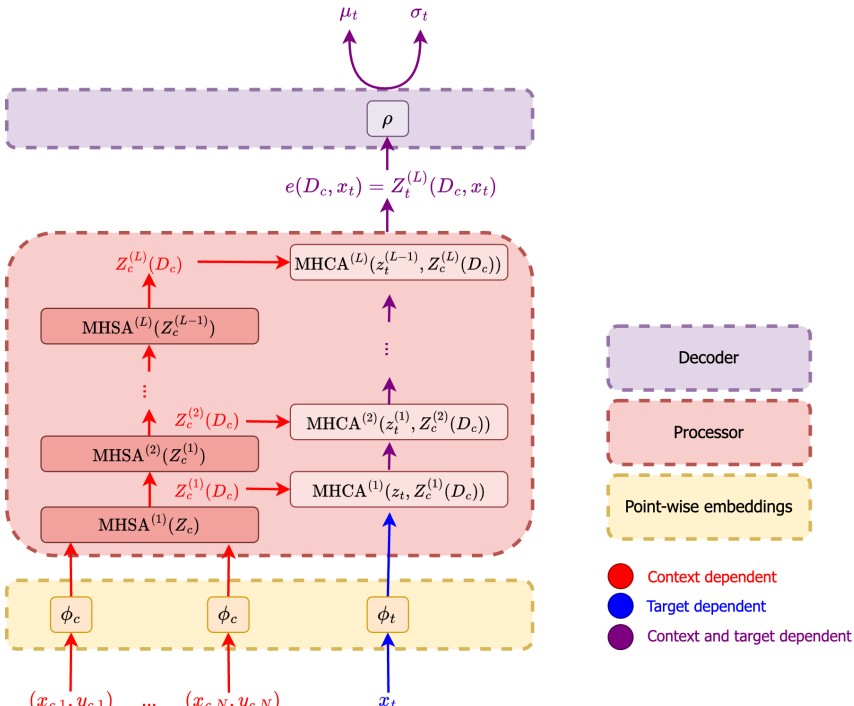

**Figure 7:** A diagram illustrating the architecture of the TNP (Nguyen and Grover, 2022).

Thus, the latent representation of the in-context datasets is a valid set function on the set of in-context datapoints.

## Appendix E. Other Choices of Architecture for the ICICL-TNP

As discussed in Section 3.2, there exist a number of possible ways to construct the ICICL-TNP. In Figure 10, we provide an alternative architecture for the ICICL-TNP than that shown in Figure 1. We also evaluated the performance of this architecture, and generally found little difference in performance.

## Appendix F. Pseudo-Code for the ICICL-TNP

Algorithm 1 shows pseudo-code for a single forward pass through the ICICL-TNP architecture shown in Figure 1.

## Appendix G. ICICL-CNP

It is possible to extend the capabilities of the CNP to perform in-context in-context learning. In addition to obtaining a latent representation of the context dataset, we obtain latent representations of the in-context datasets in an identical manner (i.e. using a Deepset (Zaheer et al., 2017)). These are aggregated, and the result is then aggregated with the context latent representation to get a latent representation of both the context dataset and in-context

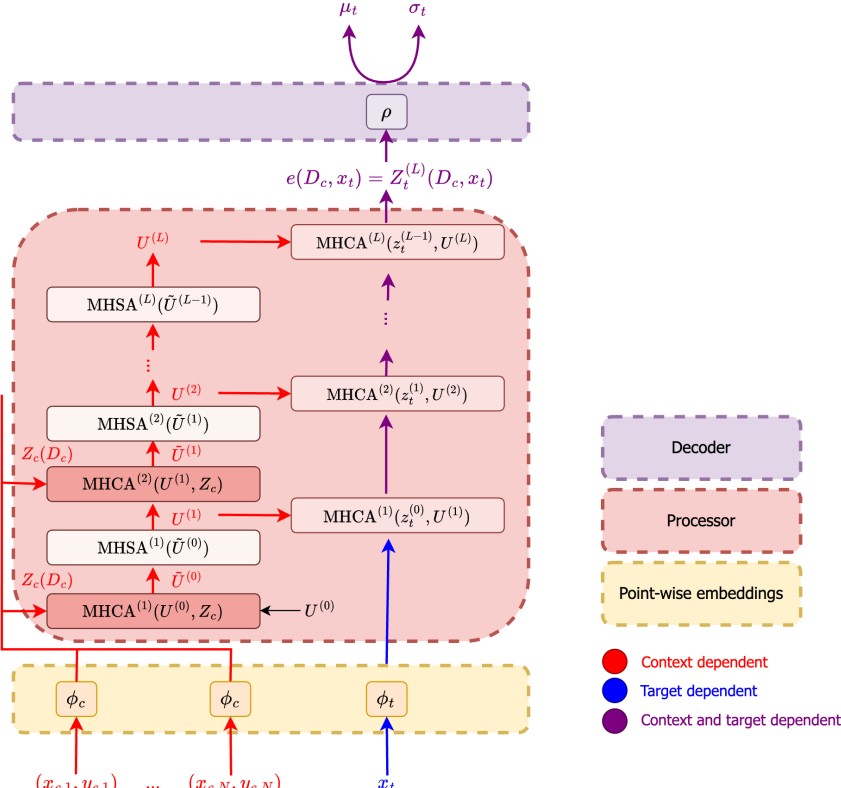

**Figure 8:** A diagram illustrating the architecture of the LBANP (Feng et al., 2022).

datasets. This is then aggregated with the latent representation of the target input, which is then fed into the decoder. We provide a diagrammatic illustration of this in Figure 11.

## Appendix H. Experiment Details

For all models, we use an embedding / token size of $D_z = 128$, and point-wise encoder and decoder consisting of an MLP with two hidden layers of dimension $D_z$. The decoder parameterises the mean and pre-softplus variance of a Gaussian likelihood with heterogeneous noise.

All transformer-based models use five layers of operations, with both the MHCA and MHSA layers using $H = 8$ heads of dimension $D_V = D_{QK} = 16$. In each of the attention blocks, we apply a residual connection consisting of layer-normalisation to the input tokens followed by the attention mechanism. Following this, there is another residual connection consisting of layer-normalisation followed by a point-wise MLP with two hidden layers of dimension $D_z$. Initial pseudo-token values are sampled from a standard normal distribution.

All CNP models use Deepsets consisting of MLPs with five layers of dimension $D_z$ for the point-wise embedding. The latent representations for each dataset are obtained by aggregating (mean) the point-wise embeddings for all the datapoints contained in that dataset. The in-context latent representation is obtained by aggregating (mean) the latent

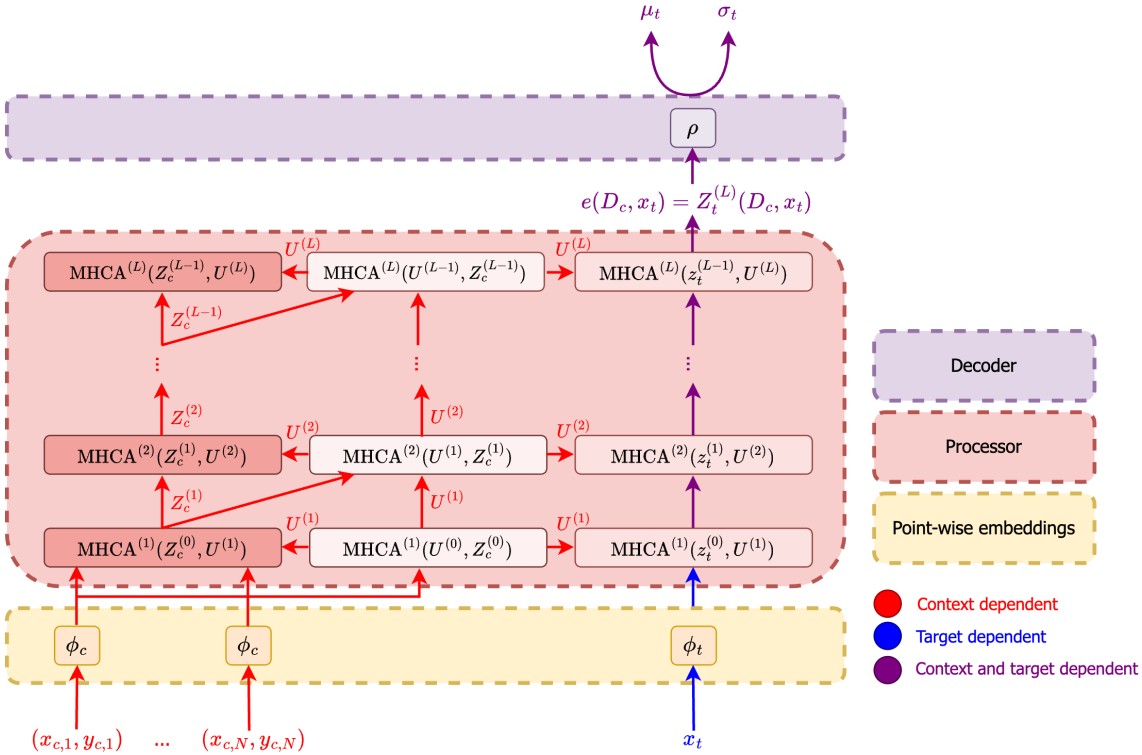

**Figure 9:** A diagram illustrating the architecture of the IST-style TNP (Lee et al., 2019).

representations for each in-context dataset. The in-context latent representation and context latent representation are aggregated through concatenation, being then concatenated again with the representation for the target location. This is then passed through a decoder, consisting of two hidden layers of dimension $D_z$.

We optimise the model parameters using AdamW (Loshchilov and Hutter, 2017) with a learning rate of $5 \times 10^{-4}$ and batch size of 16. We apply clipping to gradients with magnitudes greater than 0.5.

### H.1. Synthetic 1-D Regression

For each dataset and in-context datasets, we first randomly sample a kernel between RBF and periodic. Then, we sample the kernel's hyperparameter $\ell$ - length-scale in the case of the RBF kernel and period in the case of the periodic kernel. This is sampled according to $\log \ell \sim \mathcal{U}_{[\log 0.25, \log 4]}$. In terms of the task, we sample the number of context points $N_c \sim \mathcal{U}\{1, 64\}$, the number of in-context datasets $N_{ic} \sim \mathcal{U}\{0, 5\}$, the number of points in each in-context dataset $N_{ic,c} \sim \mathcal{U}\{64, 128\}$, the context inputs $x_{c,n} \sim \mathcal{U}_{[-2,2]}$, the target inputs $x_{t,n} \sim \mathcal{U}_{[-4,4]}$, and the inputs for the in-context datasets $x_{ic,j,n} \sim \mathcal{U}_{[-4,4]}$. All tasks use the same number of target points $N_t = 128$. The observations for each task are drawn from a GP with kernel

$$k_{\text{obs}} = k + \sigma_n^2 \delta(x - x') \tag{20}$$

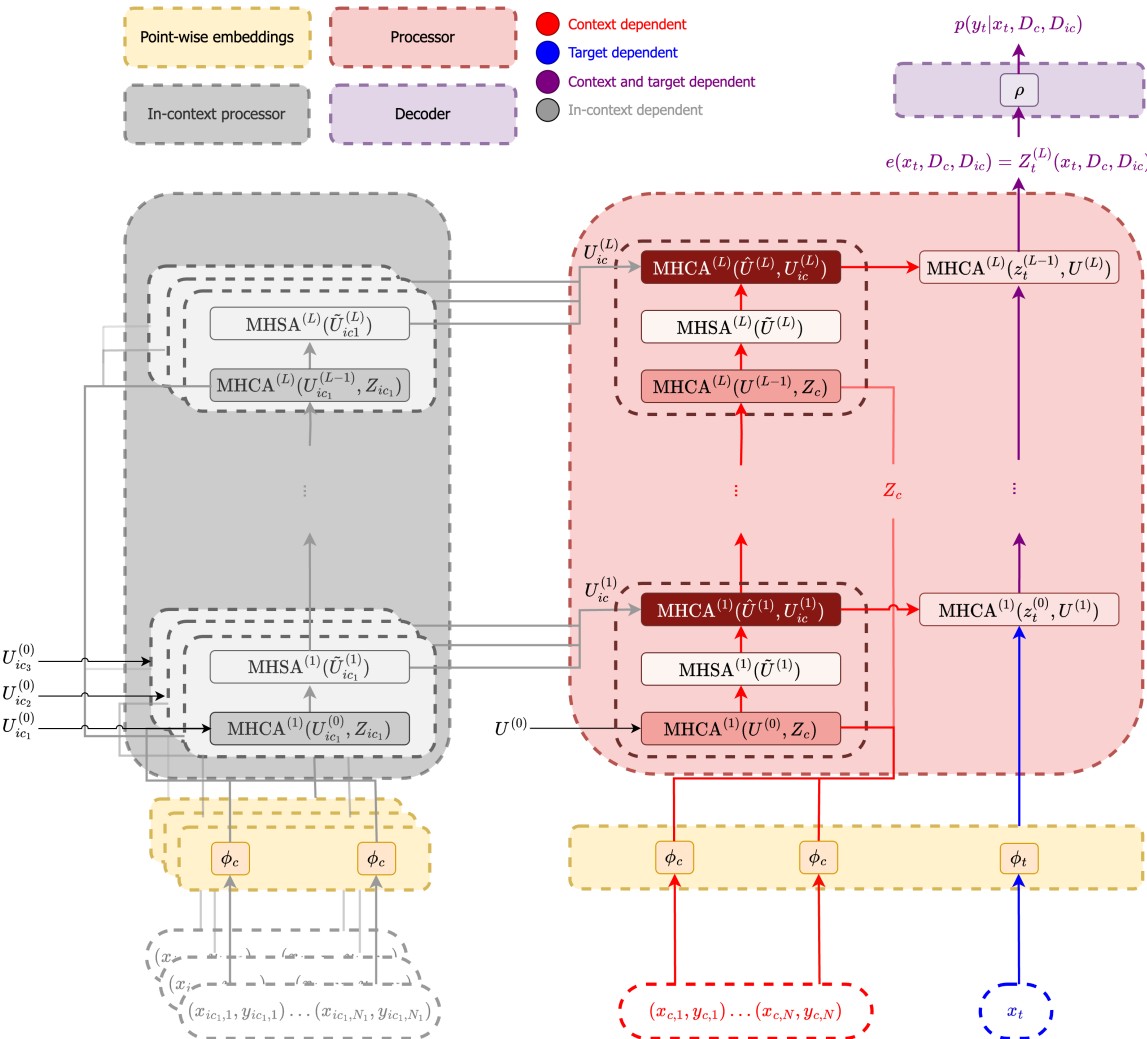

**Figure 10:** A diagram illustrating a possible choice for the architecture of the ICICL-TNP.

where the observation noise $\sigma_n = 0.2$.

For the ICICL-TNP, we use $M = 32$ context pseudo-tokens and $M_{ic} = 32$ in-context pseudo-tokens for each in-context dataset. For the PT-TNP, we use $M = 32$ pseudo-tokens. Each model is trained for 1,000 epochs, with each epoch consisting of 1,000 iterations. We evaluate the performance of each model on 80,000 test datasets.

### H.1.1. ADDITIONAL RESULTS

We also study the behaviour of the ICICL-TNP when conditioning on in-context datasets that are drawn from a different stochastic process as compared to the context dataset. In particular, in the example from fig. 2($a$), the GP the context datapoints are drawn from has a periodic kernel with $\ell = 1.85$. In this experiment, we investigate the predictive distribution in five scenarios:

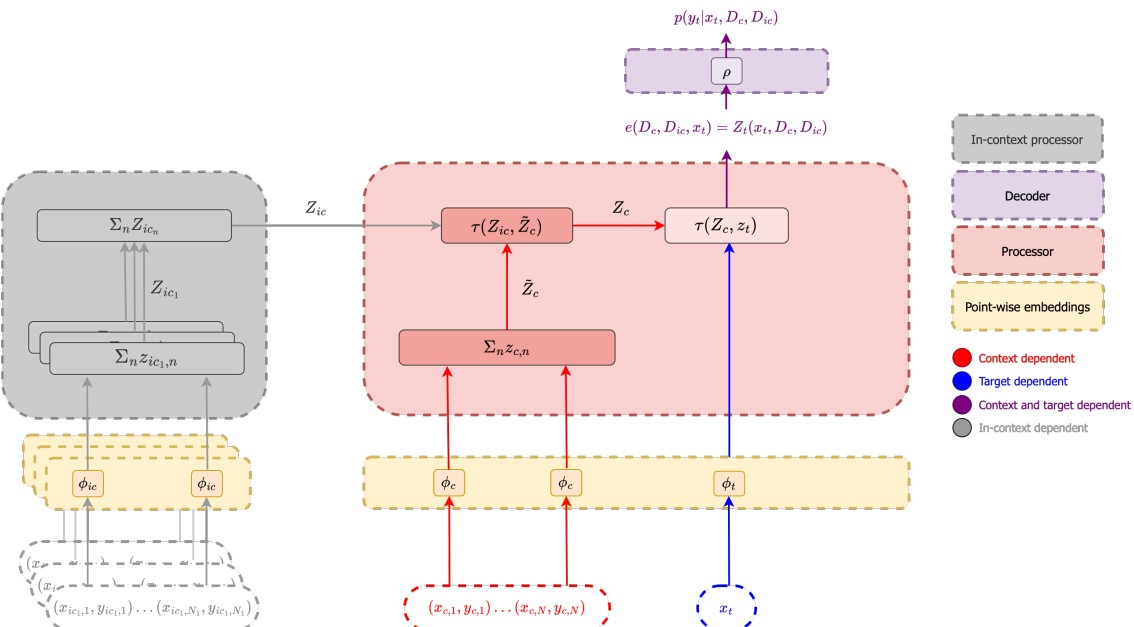

**Figure 11:** A diagram illustrating the ICICL-CNP architecture with three in-context datasets. The point-wise embedding layer is used to get an initial representation of all datapoints, including the target input location $\mathbf{x}_t$. Then, for each dataset, its latent representation is obtained using the standard Deepset approach (Zaheer et al., 2017). The latent representations for the in-context datasets are combined, and then aggregated with the latent representation for the context dataset. The resultant latent representation is aggregated with the target representation, and the processor outputs the encoder representation $e(\mathbf{x}_t, \mathcal{D}_c, \{\mathcal{D}_{ic,j}\}_{j=1}^{j=N_{ic}})$.

---

**Algorithm 1:** Forward pass through the ICICL-TNP.

---

**Input:** Context dataset $\mathcal{D}_c$, target inputs $\mathbf{x}_t$, in-context datasets $\{\mathcal{D}_{ic,j}\}_{j=1}^{N_{ic}}$, initial
context pseudo-tokens $\mathbf{U}^{(0)}$, initial in-context tokens $\mathbf{U}_{ic}^{(0)}$, pointwise-encoder $\phi$,
MHCA / MHSA parameters.

**begin**

$\quad \mathbf{z}_{c,n} \leftarrow \phi(\mathbf{x}_{c,n}, \mathbf{y}_{c,n});$

$\quad \mathbf{z}_t \leftarrow \phi(\mathbf{x}_t);$

$\quad \mathbf{z}_{ic,j,n} \leftarrow \phi(\mathbf{x}_{ic,j,n}, \mathbf{y}_{ic,j,n});$

$\quad$ **for** $\ell = 1, \ldots, L$ **do**

$\quad\quad \tilde{\mathbf{U}}^{(\ell)} \leftarrow \text{MHCA}(\mathbf{U}^{(\ell-1)}, \mathbf{Z}_c);$

$\quad\quad \hat{\mathbf{U}}_{ic,j}^{(\ell)} \leftarrow \text{MHCA}(\mathbf{U}_{ic,j}^{(\ell-1)}, \mathbf{Z}_{ic,j});$

$\quad\quad \tilde{\mathbf{U}}_{ic,j}^{(\ell)} \leftarrow \text{MHCA}(\hat{\mathbf{U}}_{ic,j}^{(\ell)}, \tilde{\mathbf{U}}^{(\ell)});$

$\quad\quad \mathbf{U}_{ic,j}^{(\ell)} \leftarrow \text{MHSA}(\tilde{\mathbf{U}}_{ic,j}^{(\ell)});$

$\quad\quad \mathbf{U}_{ic}^{(\ell)} \leftarrow \text{cat}(\mathbf{U}_{ic,1}^{(\ell)}, \ldots, \mathbf{U}_{ic,N_{ic}}^{(\ell)});$

$\quad\quad \mathbf{U}^{(\ell)} \leftarrow \text{MHCA}(\tilde{\mathbf{U}}^{(\ell)}, \mathbf{U}_{ic}^{(\ell)});$

$\quad\quad \mathbf{z}_t^{(\ell)} \leftarrow \text{MCHA}(\mathbf{z}_t^{(\ell-1)}, \mathbf{U}^{(\ell)});$

$\quad$ **end**

$\quad$ **return** $p(\cdot|\rho(\mathbf{z}_t^{(L)}));$

**end**

---

1. No in-context datasets;

2. Three in-context datasets that are drawn from the same stochastic process as the context datapoints;

3. Three in-context datasets where the datapoints are generated from a GP with an RBF, instead of a periodic kernel (with $\ell = 0.52$);

4. Three in-context datasets where the datapoints are generated from a GP with a periodic kernel and a similar (but not identical) period $\ell = 1.76$;

5. Three in-context datasets where the datapoints are generated from a GP with a periodic kernel and a significantly different period $\ell = 0.28$.

The resulting predictive distributions are shown in Figure 12. As already noted in the main text, when conditioning on in-context data drawn from the same stochastic process, as opposed to no in-context information, the predictions improve significantly and the uncertainty decreases. When the in-context and context datapoints are drawn from a GP with a different kernel (RBF vs. periodic), the predictions of the ICICL-TNP are more uncertain, especially far away from the data, but the model still manages to fit the data well in regions with high context datapoints concentration (Figure 12(c)). When the in-context and context datapoints are drawn from a GP with the same type of kernel and a similar period, the predictions improve significantly as compared to when no in-context information is available

(Figure 12(d)). This suggests that even if the two stochastic processes are not exactly the same, but with similar characteristics, the model is still able to extract useful information from the in-context data. In contrast, when the two stochastic processes have wildly different characteristic (i.e. period), the model is unable to explain the data (Figure 12(e)), leading to a large increase in uncertainty in all regions. We note that this could be used as a way to identify whether the in-context and context datasets are indeed generated by the same stochastic process, with the ICICL-TNP outputting uncertain predictions when there is a severe discrepancy.

### H.1.2. Results for alternative architecture

We also provide the results on this task for the alternative architecture shown in Figure 10. We did not observe a significant difference between this model (ICICL-TNP2) and ICICL-TNP on this task, as illustrated in Table 4.

| $N_{ic}$ | ICICL-TNP | ICICL-TNP2 |
|---|---|---|
| 0 | $-0.607 \pm 0.005$ | $-0.612 \pm 0.005$ |
| 1 | $-0.499 \pm 0.005$ | $-0.491 \pm 0.005$ |
| 2 | $-0.474 \pm 0.005$ | $-0.473 \pm 0.005$ |
| 3 | $-0.469 \pm 0.005$ | $-0.467 \pm 0.005$ |
| 4 | $-0.467 \pm 0.005$ | $-0.464 \pm 0.005$ |
| 5 | $-0.466 \pm 0.005$ | $-0.463 \pm 0.005$ |

**Table 4:** Comparison of test log likelihood for the ICICL-TNP and the ICICL-TNP2 architectures on the synthetic GP task for a varying number of in-context datasets ($N_{ic}$). The two models have similar performance (within one standard deviation).

### H.1.3. Results for OOD testing

We present some additional examples of predictions on samples that are drawn outside of the distribution the models have been trained on. In the top row of Figure 13 we consider samples drawn from a GP with RBF kernel and $\ell = 5.19$, whereas in the bottom row we consider the challenging task of modelling datapoints drawn from a GP with a periodic kernel and $\ell = 0.24$. In both cases the ICICL-TNP is conditioned on three in-context datasets, containing datapoints satisfying the same stochastic process as the context data. The ICICL-TNP manages to leverage in-context learning to improve its predictions even when the samples are drawn from a different distribution than the one the model has been trained on.

## H.2. Image Completion

For each dataset and in-context datasets, we first sample a label $\ell \sim \mathcal{U}\{0, 9\}$, and then sample images from the subset of MNIST images that correspond to that label. The number of context points is sampled as $N_c \sim \mathcal{U}\{N/100, N/5\}$, the number of in-context datasets is sampled as $N_{ic} \sim \mathcal{U}\{0, 3\}$, and the number of datapoints for each of the in-context datasets

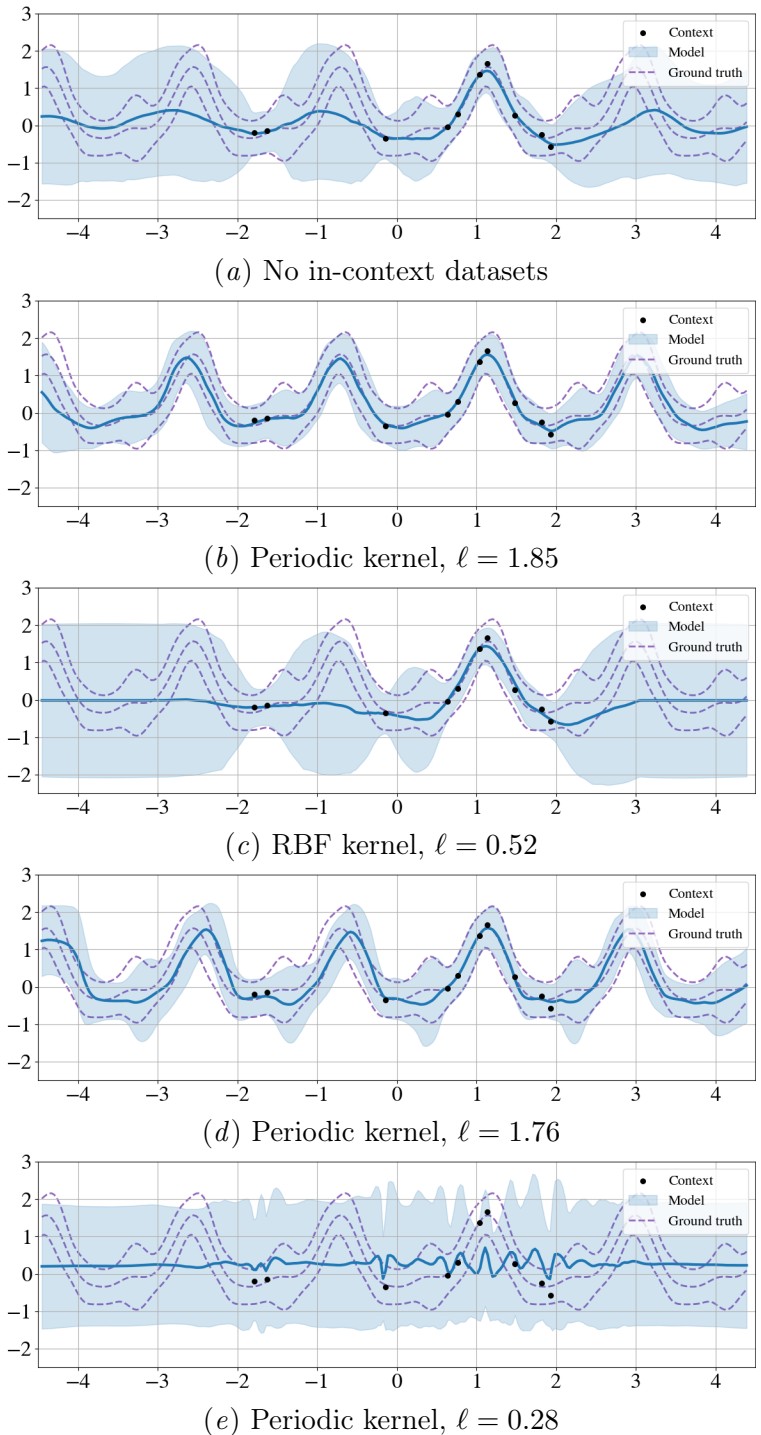

(a) No in-context datasets

(b) Periodic kernel, $\ell = 1.85$

(c) RBF kernel, $\ell = 0.52$

(d) Periodic kernel, $\ell = 1.76$

(e) Periodic kernel, $\ell = 0.28$

**Figure 12:** A comparison between the predictive distribution of the ICICL-TNP with different in-context conditioning information. The GP kernel used to generate the context datapoints was periodic with $\ell = 1.85$.

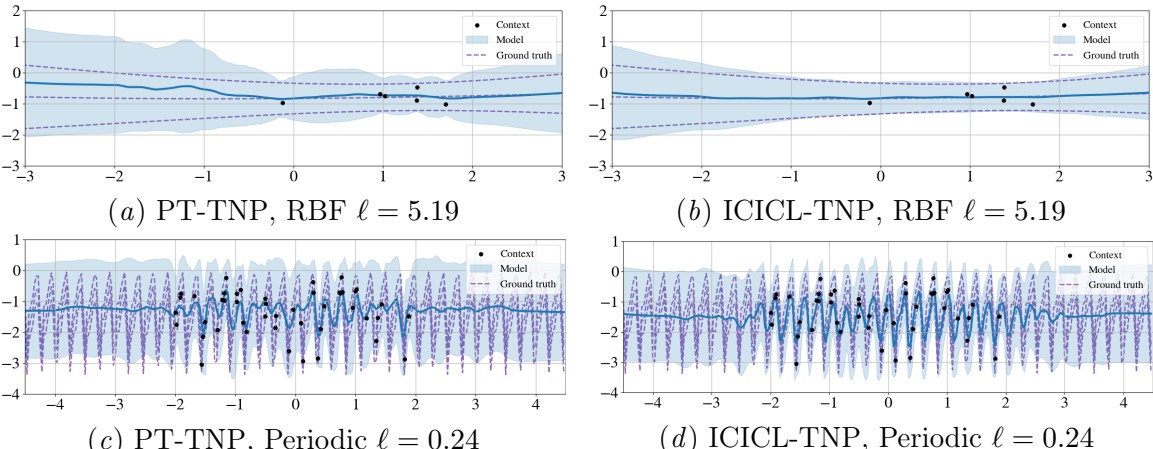

(a) PT-TNP, RBF $\ell = 5.19$        (b) ICICL-TNP, RBF $\ell = 5.19$

(c) PT-TNP, Periodic $\ell = 0.24$      (d) ICICL-TNP, Periodic $\ell = 0.24$

**Figure 13:** A comparison between the predictive distribution when tested OOD of the PT-TNP (left) and the ICICL-TNP (right). In the top row the samples come from a GP with RBF kernel and $\ell = 5.19$. In the bottom row the samples come from a GP with periodic kernel and $\ell = 0.24$. The ICICL-TNP is conditioned on three in-context datasets.

is sampled as $N_{ic,j} \sim \mathcal{U}\{N/100, N/2\}$. The number of target points is set to the remaining pixels in the image (i.e. $N_t = N - N_c$).

For the ICICL-TNP, we use $M = 64$ context pseudo-tokens and $M_{ic} = 64$ in-context pseudo-tokens for each in-context dataset. For the PT-TNP, we use $M = 64$ pseudo-tokens. Each model is trained for 500 epochs, with each epoch consisting of 625 iterations with a batch size of 16. We evaluate the performance of each model on 2,500 images (the test dataset split into groups of four, so that each group has one context dataset and three in-context datasets, so that a maximum of three is available for use).

### H.2.1. ADDITIONAL RESULTS

In Figure 14 we compare the predictive distribution of the ICICL-TNP when conditioning on in-context datasets drawn from different stochastic processes (MNIST labels).

### H.3. Environmental Data

The environmental dataset consists of surface air temperatures derived from the fifth generation of the European Centre for Medium-Range Weather Forecasts (ECMWF) atmospheric reanalyses (ERA5) (Copernicus Climate Change Service, 2020). The data has a latitudinal and longitudinal resolution of 0.5°, and temporal resolution of an hour. We consider data collected in 2019, sub-sampled at a temporal resolution of six hours. The dataset consists of data within the latitude / longitude range of [42°, 53°] / [8°, 28°] (roughly corresponding to central Europe), with the training data corresponding to the first six months of 2019, and the test data corresponding to the last six month. Individual datasets are obtained by sub-sampling the larger region, with each dataset consists of a $[10, 10, 3]$ grid spanning 5° across each axis and 18 hours. We also provide surface elevation as additional inputs, such that $D_x = 4$. The inputs and outputs are standardised using the mean and standard

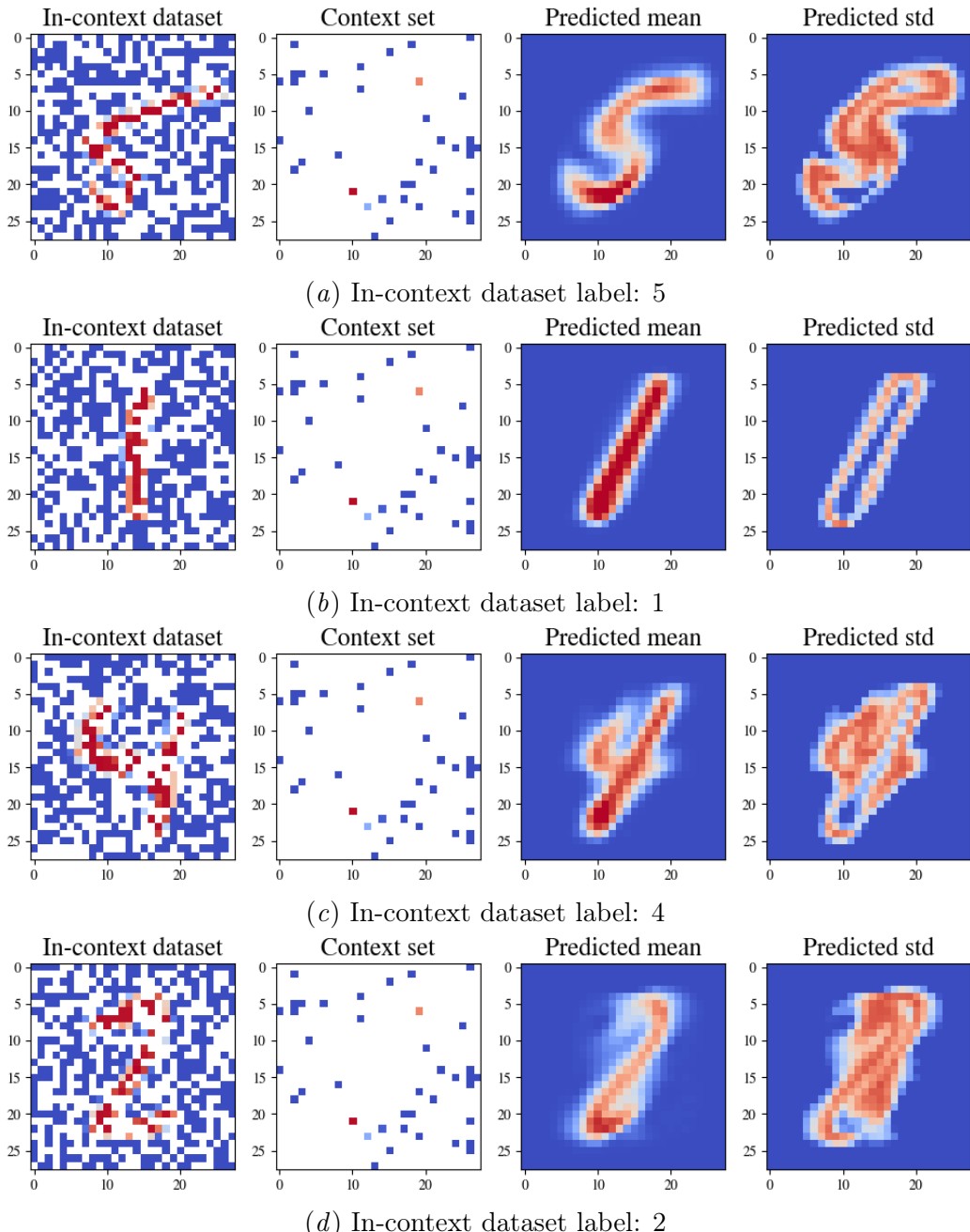

**Figure 14:** A comparison between the predictive distribution of the ICICL-TNP when conditioning on in-context datasets sampled from different MNIST labels. Observe that when the context set is not compatible with the label of the in-context dataset (i.e. 4 and 2), the predictive distribution is less confident.

deviation values obtained from data within the training region. Each dataset consists of a maximum of $N = 300$ datapoints, from which the number of context points are sampled according to $N_c \sim \mathcal{U}\{N/100, N/3\}$, with the remaining set as target points. The number of in-context datasets is sampled as $N_{ic} \sim \mathcal{U}\{0, 2\}$, and the number of datapoints in each of the in-context datasets is sampled as $N_{ic,j} \sim \mathcal{U}\{N/3, N\}$.

For the ICICL-TNP and PT-TNP, we use $M = 32$ context pseudo-tokens and $M_{ic} = 32$ in-context pseudo-tokens for each in-context dataset. We train each model for 500 epochs, with each epoch consisting of 1,000 iterations with a batch size of 16.

As opposed to the other experiments, in the case of the environmental data the distinction between the context and in-context datasets is less clear, and is a result of the experiment design choices. In particular, in this experiment we chose to select both the context and in-context data from the same spatial region, but from non-overlapping time ranges. However, we do not pass any time information to the model. Thus, we designed the experiment in order to capture an instance in which ICICL would be beneficial (e.g. if we did not have access to the absolute time of the context / in-context samples).

In order to better understand the benefits of ICICL, for this dataset we also included an additional baseline—PT-TNP-merged—with the results shown in Table 5. The PT-TNP-merged is trained and tested with an expanded context dataset (as opposed to PT-TNP), consisting of the union between the original context dataset and the in-context datasets (i.e. $\mathcal{D}'_c = \mathcal{D}_c \cup \{\mathcal{D}_{ic,j}\}_{j=1}^{N_{ic}}$). The number within the brackets from Table 5 indicates the number of in-context datasets that have been appended to the context dataset during testing. Note, once again, that in this setup we are assuming we do not have access to the absolute time of the samples, and hence do not include time as a variable in the model.

As shown in Table 5, as opposed to ICICL-TNP, PT-TNP-merged does not show a gain in performance by extending the context set with the in-context data. We hypothesise this is because of the difficulty of the task, which involves learning complex relationships across long periods of time. Moreover, whenever in-context data is provided, PT-TNP-merged underperforms the ICICL-TNP model, suggesting that ICICL leads to more robust results when one has access to additional data (i.e. in-context data) that shares similarities with the data of interest (i.e. context data), but is perhaps not complete enough (e.g. missing absolute temporal information) to naively treat it in the same way as the context set.

| Model | Test Log-Likelihood |
|---|---|
| PT-TNP | $1.15 \pm 0.01$ |
| PT-TNP-merged (0) | $1.14 \pm 0.01$ |
| PT-TNP-merged (1) | $1.15 \pm 0.01$ |
| PT-TNP-merged (2) | $1.14 \pm 0.01$ |
| ICICL-TNP (0) | $1.15 \pm 0.01$ |
| ICICL-TNP (1) | $1.18 \pm 0.01$ |
| ICICL-TNP (2) | $1.19 \pm 0.01$ |

**Table 5:** Comparison of the test log-likelihood on the environmental data for the PT-TNP and ICICL-TNP with varying number of in-context datasets (indicated within brackets). As an additional baseline, we also consider the PT-TNP-merged where the in-context datasets are merged into the context datasets.

