# OpenReview forum: "In-Context In-Context Learning with Transformer Neural Processes"
_approximateinference.org/AABI/2024/Symposium_Archival_Track — AABI 2024 - Archival Track_

### Official Review · Reviewer_hMZt · 2024-04-24
**Interesting formulation of in-context learning**

**Rating:** 7
**Confidence:** 3

**Review:**

In this paper, the authors propose to do in-context in-context learning by using pseudo-token transformer neural processes to use information from multiple different datasets in-context in addition to the usual in-context information for making predictions. Their model complexity scales with the number of pseudo-tokens rather than tokens, which makes such an approach tractable. They also show that using this additional information can lead to improvements, both theoretically and empirically. In the experiments, ICICL-TNP does no worse than plain PT-TNP when no additional in-context datasets are available. But with additional in-context datasets, it does better.

Overall, the work is very interesting and novel to my knowledge. The paper is mostly well written and readable, and being able to improve performance by using multiple datasets in-context when available sounds like a very useful thing to have.

The pseudo-token approach is useful to mitigate computational complexity in any transformer architecture, and is not specific to this method. The main contribution here seems to be to make use of it in this specific way (multiple datasets, neural processes).

One clarity concern is that Fig. 1 is very hard to understand. There are lots of lines that are overlapping, and some lines such as the one going from the IC dataset to the “Process” (red colour) take a seemingly circuitous route to the first MHCA layer.  There is also a large number of acronyms used in the paper which makes the reading a bit harder.

It would also have been nice to see additional scaling experiments, since scalability of the method is one of the main claims made in the beginning.

Overall, it’s a solid paper, and I recommend accept.

---

### Official Review · Reviewer_RiH8 · 2024-04-25
**Exaggerated limitations and unfair comparisons to existing methods**

**Rating:** 5
**Confidence:** 3

**Review:**

### Summary:
The authors propose in-context in-context learning pseudo-token transformer neural processes (ICICL-TNP) to address a shortcoming of existing pseudo-token transformer neural processes (PT-TNP) that they can only use one context dataset but cannot leverage additional potentially beneficial datasets (in-context datasets as referred to in the manuscript). ICICL-TNP uses cross attention to integrate information from the in-context datasets. The authors theoretically analyze and empirically demonstrate the advantages of ICICL-TNP against PT-TNP.

### Strengths:
The authors conduct theoretical analysis and extensive experiments to demonstrate the advantage of ICICL-TNP.

### Weaknesses:
1. The authors overstate the limitations of existing PT-TNP methods. In Section 3.1, the definition of ICICL, the authors assume that the additional in-context datasets $\mathcal{D}_j$ are drawn from the same distribution $P(\xi_i)$ that the context dataset $\mathcal{D}_i$ is drawn from. This can be handled by PT-TNP by merging additional in-context datasets into the context dataset to be a larger context dataset as they are drawn from the same distribution $\mathcal{D}'_i = \mathcal{D}_i \cup \mathcal{D}_j$.
2. Unfair comparison to baseline methods. Given the above method, for a fair comparison, the baseline for ICICL-TNP (1) should not only be PT-TNP, but PT-TNP with $\mathcal{D}'_i = \mathcal{D}_i \cup \mathcal{D}_j$, the baseline for ICICL-TNP (2) should be PT-TNP with $\mathcal{D}'_i = \mathcal{D}_i \cup \mathcal{D}_j \cup \mathcal{D}_k$, etc.

### Suggestions for Presentation Improvements
In the Introduction, the authors discuss using PDE-simulated Navier-Stokes equations to motivate their work but fail to clarify the concept of the Reynolds number, which might confuse readers. In Section 3.2, terms such as "multi-head cross attention" (MHCA) are only defined in figure captions and then mentioned in the main text, making it difficult for readers to understand the reference to MHCA. Additionally, the manuscript contains several typographical and grammatical errors. For instance, at the end of Page 3, the sentence reads, "However, this computational complexity can be addressed through the use of pseudo-token based transformers, which reduce the computational complexity of the standard transformer through the use of pseudo-tokens." This sentence is redundant and could be more concise.

---

### Official Review · Reviewer_1reL · 2024-04-26
**In this paper, the authors present an approach under the framework of neural process to consider extra information (termed as “in-context”) besides context points in the prediction for target points. This approach is based on self-attentions within in-context and context datasets and cross-attentions between the two. Like previous works, pseudo-tokens are employed to reduce the computational cost. The evaluation was done on synthetic datasets and two semi-synthesized datasets based on real world datasets.**

**Rating:** 5
**Confidence:** 3

**Review:**

Strength

The problem to solve seems interesting and a proper solution is needed.

The reported experiment results are supportive to the effectiveness of the proposed method in taking advantage of additional datasets.


Weakness

In the image completion experiment, the authors used images of the same label to form in-context datasets. In practice, it is likely that one would not know the label of the image to be completed as demonstrated in Figure 5. It would be interesting to see when images with different labels used for in-context datasets, how the proposed method performs.

In the design of the environmental data experiment, it seems that the proposed method is simply to have more data to base upon for predictions. I wonder what the difference between in-context and context data really is. How about simply treat in-context as context and combine all context to run PT-TNP? The comparison with this result could help to show whether the proposed architecture is really needed for this application.

In general, the paper can be improved with more experiments on real world problems.

---

### Official Review · Reviewer_qeCT · 2024-04-26
**Well-written extension to the neural processes literature**

**Rating:** 7
**Confidence:** 4

**Review:**

The paper formulates a task setup called in-context in-context learning, where a model seeks to learn the posterior prediction map from context sets to the posterior predictive distribution over target outputs while additionally conditioned on other in-context datasets. The paper proposes a transformer-based architecture for modeling this task, ICICL-TNP, where permutation invariance is preserved with respect to points within the context dataset and in-context datasets, as well as across in-context datasets. The paper presents experiments on 1. a synthetic regression task based on samples from Gaussian processes, and 2. an image completion task, demonstrating empirically the benefit of taking into account the additional in-context datasets, matching the theory.

Strengths:
1. Very clear and well-written. Background section and setup were especially nice to read / easy to follow with clear notation.
2. The proposed architecture is novel and exhibits desirable permutation invariance properties.
3. Comparisons between the formulated task setup and other tasks / previous work are well-articulated.

Weaknesses:
1. Lack of real-world applications considered / discussed. Outside of learning low-dimensional functions, what settings would benefit from / be able to take advantage of the task setup proposed? The authors mention in the conclusion that one potential application would be "image generation tools which can be provided with additional images that the user wishes the generated sample to be similar to," but even in that example it is not clear why the two levels of in-context are useful. For instance, in the image example in the experiments, the "dataset" is simply a full image (function of position to pixel value) and the inner "in-context learning" loop is simply infilling; then, what the current paper calls in-context in-context learning would probably be what previous literature refers to as just in-context learning (i.e., feed in examples of partial images and their completions before asking the model to output the completion of a new image). Some discussion of this comparison would be useful, as would providing some concrete real-world examples where the two levels of context is available / practical. I do think that one contribution of this work even beyond unlocking new applications is satisfying exchangeability of the in-context datasets in the modeling choices; this is a point that could be raised when comparing settings where the paper's in-context in-context learning fits the task description of in-context learning (one level of in-context, given a different view of what is an example vs. dataset).

---

### Official Review · Reviewer_sLxU · 2024-04-26
**This paper extends in-context learning to multiple datasets instead of multiple observations**

**Rating:** 7
**Confidence:** 4

**Review:**

This paper considers the problem of in-context in-context learning (ICICL). Unlike standard in-context learning, which assumes access to a set of observed datapoints, ICICL assumes access to a set of similar observed datasets. A specific transformer neural process-based architecture called the ICICL-TNP is proposed, which utilizes multi-head cross attention to access information from the related datasets.

While ICL has received a lot of attention, the problem of ICICL is novel to my knowledge. It seems likely to be a useful direction, as it provides the practitioner an additional way to specify knowledge about a task that does not require fine-tuning. The main claims of the paper, namely (a) that the ICICL-TNP performs similarly to the PT-TNP when there is no in-context dataset and (b) that the ICICL-TNP improves in performance with access to related datasets, are both supported with empirical evidence across three datasets.

Overall, this is a well-written paper that pushes in an interesting direction and has convincing results. The experiments are conducted on relatively small-scale datasets, but even at this scale the improvements of the proposed architecture are evident. The related work could be expanded to include other work from the pre-transformer era that conditions on multiple datasets, such as [1].

[1] Edwards, Harrison, and Amos J. Storkey. “Towards a Neural Statistician.” ICLR 2017.

---

### Meta-Review · Area_Chair_tBmW · 2024-05-24

**Recommendation:** Accept (Poster)
**Confidence:** 4

**Metareview:**

This paper introduces in-context in-context learning (ICICL). While standard in-context learning assumes access to a set of observed datapoints, ICICL leverage additional related datasets to improve predictions. The paper proposes a specific transformer neural process-based architecture (ICICL-TNP) and demonstrates that the model outperforms standard models, especially when additional in-context datasets are available.

The reviewers agree that the paper is well-written, that it addresses an interesting and relevant problem, and that the experiments show convincing results. There were some concerns that the experiments were conducted on small-scale datasets and that the real-world applications were not considered / discussed. These concerns were discussed during rebuttal, and I encourage the authors to incorporate the main points of their responses into the paper and when presenting the work.

Overall, this looks like novel work, which clearly presents an effective approach to using multiple datasets for improved predictions.

---

### Decision · Program_Chairs · 2024-05-27

Accept